# Inhibiting WNT and NOTCH in renal cancer stem cells and the implications for human patients

Annika Fendler[1,2,3,4], Daniel Bauer[1], Jonas Busch[2,3], Klaus Jung [3,4], Annika Wulf-Goldenberg[5], Severine Kunz [6], Kun Song[7], Adam Myszczyszyn[1], Sefer Elezkurtaj[8], Bettina Erguen [3], Simone Jung[1], Wei Chen [2,7,9] & Walter Birchmeier[1,2✉]

Current treatments for clear cell renal cell cancer (ccRCC) are insufficient because two-thirds of patients with metastases progress within two years. Here we report the identification and characterization of a cancer stem cell (CSC) population in ccRCC. CSCs are quantitatively correlated with tumor aggressiveness and metastasis. Transcriptional profiling and single cell sequencing reveal that these CSCs exhibit an activation of WNT and NOTCH signaling. A significant obstacle to the development of rational treatments has been the discrepancy between model systems and the in vivo situation of patients. To address this, we use CSCs to establish non-adherent sphere cultures, 3D tumor organoids, and xenografts. Treatment with WNT and NOTCH inhibitors blocks the proliferation and self-renewal of CSCs in sphere cultures and organoids, and impairs tumor growth in patient-derived xenografts in mice. These findings suggest that our approach is a promising route towards the development of personalized treatments for individual patients.

[1] Cancer Research Program, Max Delbrueck Center for Molecular Medicine (MDC) in the Helmholtz Association, Berlin, Germany. [2] Berlin Institute of Health (BIH), Berlin, Germany. [3] Department of Urology, Charité-University Medicine, Berlin, Germany. [4] Berlin Institute for Urologic Research, Berlin, Germany. [5] Experimental Pharmacology and Oncology GmbH (EPO), Berlin, Germany. [6] Electron Microscopy Core Facility, Max Delbrueck Center for Molecular Medicine (MDC) in the Helmholtz Association, Berlin, Germany. [7] Berlin Institute for Medical Systems Biology (BIMSB), Max Delbrueck Center for Molecular Medicine (MDC) in the Helmholtz Association, Berlin, Germany. [8] Department of Pathology, Charité-University Medicine, Berlin, Germany. [9] Department of Biology, Southern University of Science and Technology, Shenzhen, China. ✉email: wbirch@mdc-berlin.de

Kidney cancer is the twelfth most common malignancy in the western world, and clear cell renal carcinoma (ccRCC, also called KIRC) is its most common form, affecting 70–80% of patients[1]. Currently the 5-year survival rate of localized ccRCC lies at 65%, but it drops to 10–20% once the cancer has metastasized[1]. Large scale sequencing has pointed to loss of the short arm of chromosome 3 and mutations of the Von Hippel Lindau (VHL) gene as the main drivers of ccRCC. Mutations of PBRM1, SETD2, BAP1 are found at lower rates[2,3].

The heterogeneity observed in kidney tumors has been an obstacle to successful treatment and might be a major contributor to relapse[4]. Significant improvements in post-surgical treatment have been made in the last two decades: inhibitors of multiple tyrosine kinases, of mTOR or monoclonal antibodies against VEGF[5,6]. Sequential treatments with these inhibitors improve patient outcomes; nevertheless, within 2 years most tumors progress. A more recent approach enhances immune responses to kidney tumors through checkpoint inhibitors which block PD-1 or CTLA-4 on T-cells[7], with long-lasting effects for a subset of patients. Ultimately, improving the long-term prognosis ccRCC will require personalized treatment strategies specific to the biology of each tumor.

CSCs have been characterized in many cancers and implicated in resistance to treatment, tumor recurrence, and metastatic spread; the situation in kidney cancer has been unclear[8–10]. Organoid cultures, grown from stem cells in the presence of specific growth factor cocktails, have been derived from a range of tissues and are crucial models in the investigation and treatment of a range of cancers[11]. Colon cancer organoids are being used to study the effects of pathway inhibitors and anti-cancer drugs[12]. Yet organoids derived from kidney tumors have only recently been described; here we report a well-characterized organoid model from human primary ccRCCs.

In addition, patient-derived xenografts (PDXs) derived through transplantations of cells and disease tissues into immune-compromised mice have been used as models to study renal carcinogenesis[13,14]. The fidelity that is maintained through re-passaging makes it possible to produce animals whose tumors replicate that of an individual patient and can be used to search for effective treatments.

In combination, PDX and organoids have surpassed the restrictions of working solely in immortal cell lines and animal models and permit studying response to therapies in individual tumors. Based on the behavior of any of these models, robust predictions about likely outcomes in patients can be made.

We here develop procedures to isolate CSCs from ccRCCs and analyze them through expression profiling and single-cell sequencing. We use CSCs from the tumors to produce three model systems—non-attached sphere cultures, 3D organoids, and PDX tumors—to overcome the limitations imposed by single model systems. We treat each model with small molecule inhibitors that target WNT and NOTCH at different stages. This combined approach may be a promising route toward the development of personalized treatments for individual patients leading to early phase clinical trials.

## Results

**Frequency of CSCs correlates with aggressiveness of ccRCC.** We isolated single cells from patient ccRCC tissues (labeled ccRCC1, 2 etc.) obtained during surgery (see Supplementary Table 1 for the characterization of patients) and investigated cell surface markers on their own and in combination using FACS, aiming to identify a ccRCC cell stem cell population. The selected surface markers have been previously identified as stem cell markers in the kidney (i.e. CD24, CD29, CD133)[15], cancer stem cell markers in other malignancies (CD24, CD29, Epcam, CD44, MET, CD90, ALDH1A1 activity)[16–21], and in the kidney (CD133, CD24, CD105, CXCR4)[8,9,15,22]. FACS revealed a distinct population of CXCR4+MET+ cells in patient's tumor which could be further sorted into CD44+ and CD44− cells (Fig. 1a and Supplementary Fig. 1a). The chemokine receptor CXCR4 and the receptor tyrosine kinase MET had been associated with ccRCC in previous studies[23–26]. We found that CD44, a frequent marker of CSCs[8,9,27], can further refine this population. CXCR4+ MET+CD44+ cells amounted to 2.2% of total tumor cells on average (range: 0.2–11%). We seeded FAC-sorted cells in decreasing numbers to test their sphere-forming abilities, as a read-out for the self-renewing capacity. Triple-positive cells (red) were able to form spheres at cell numbers ten-fold less than CXCR4−MET−CD44− cells (black) (Fig. 1b). Spheres formed by triple-positive cells were also larger than spheres generated from negative cells (Fig. 1c, d).

We used the same procedure to analyze the remaining markers: CD24, CD29, CD105, CD133, CD90, and ALDH1A1. CD24 and CD29 were detected almost ubiquitously on ccRCC cells (Supplementary Fig. 1b) and were therefore excluded. The other markers failed to enrich for self-renewing cells (Supplementary Fig. 1c–f). Epcam, a marker for epithelial and carcinoma cells used in other solid malignancies[28,29], was detected at low levels in ccRCC, as determined both by FACS and immunofluorescence (Supplementary Fig. 1g, h), and was likewise excluded.

To examine whether the frequency of CXCR4+MET+CD44+ cells correlated with the aggressiveness of the tumors, we used FACS to isolate cells from the primary ccRCC tumors of 41 patients selected from the 55 in Supplementary Table 1. CXCR4+ MET+CD44+ cells were highly enriched in tumors with a high pathological stage (Fig. 1e), a high Fuhrman grade (Fig. 1f)[30], venous and lymphatic invasion (Fig. 1g, h) and distant metastases (Fig. 1i) at the time of surgery. Tumor cell proliferation was estimated by scoring Ki-67 positive cells from 0 (no positive cells) to 3 (mostly positive). The frequency of CXCR4+MET+CD44+ cells was not significantly associated with Ki-67 scores (Supplementary Fig. 1i).

We also established PDX through subcutaneous transplantation of tumor cells into immune-compromised mice. PDX grow from about 15% of ccRCCs, but take rates were higher from primary tumors of patients with metastases[13]. This prompted us to establish xenografts from primary tumor tissue if distant metastases were present (Fig. 2a). With this procedure we established four PDX from eight patients. Cells from three subcutaneous tumors were triple-sorted (Supplementary Fig. 2a) and re-transplanted orthotopically into the renal parenchyma to test their tumor-initiating potential. Low numbers of CXCR4+ MET+CD44+ cells initiated tumor growth, while negative samples, with one exception, failed to do so (Fig. 2b). To examine whether CXCR4+MET+CD44+ cells produced heterogeneous tumors, we compared the orthotopic xenografts with the parental subcutaneous tumors. The histological diversity of the parental xenografts was re-established in the orthotopic xenografts (Fig. 2c). Immunofluorescence showed that PDXs were positive for both CA9 and CD10 (Supplementary Fig. 2b), confirming their ccRCC identity. Ki-67 scores in subcutaneous or orthotopic PDX were similar to or higher than in corresponding primary tumors (Supplementary Fig. 2c).

Immunofluorescence for CXCR4, MET, and CD44 confirmed that CXCR4+MET+CD44+ cells are rare in ccRCC tumors and subcutaneous PDX (Fig. 2d). Their low levels in orthotopic tumors indicated that transplanted CXCR4+MET+CD44+ cells differentiated and lost marker expression during tumor formation. We observed no preferential location of CXCR4+MET+CD44+ cells within the tumors, even though there were marked intra- and

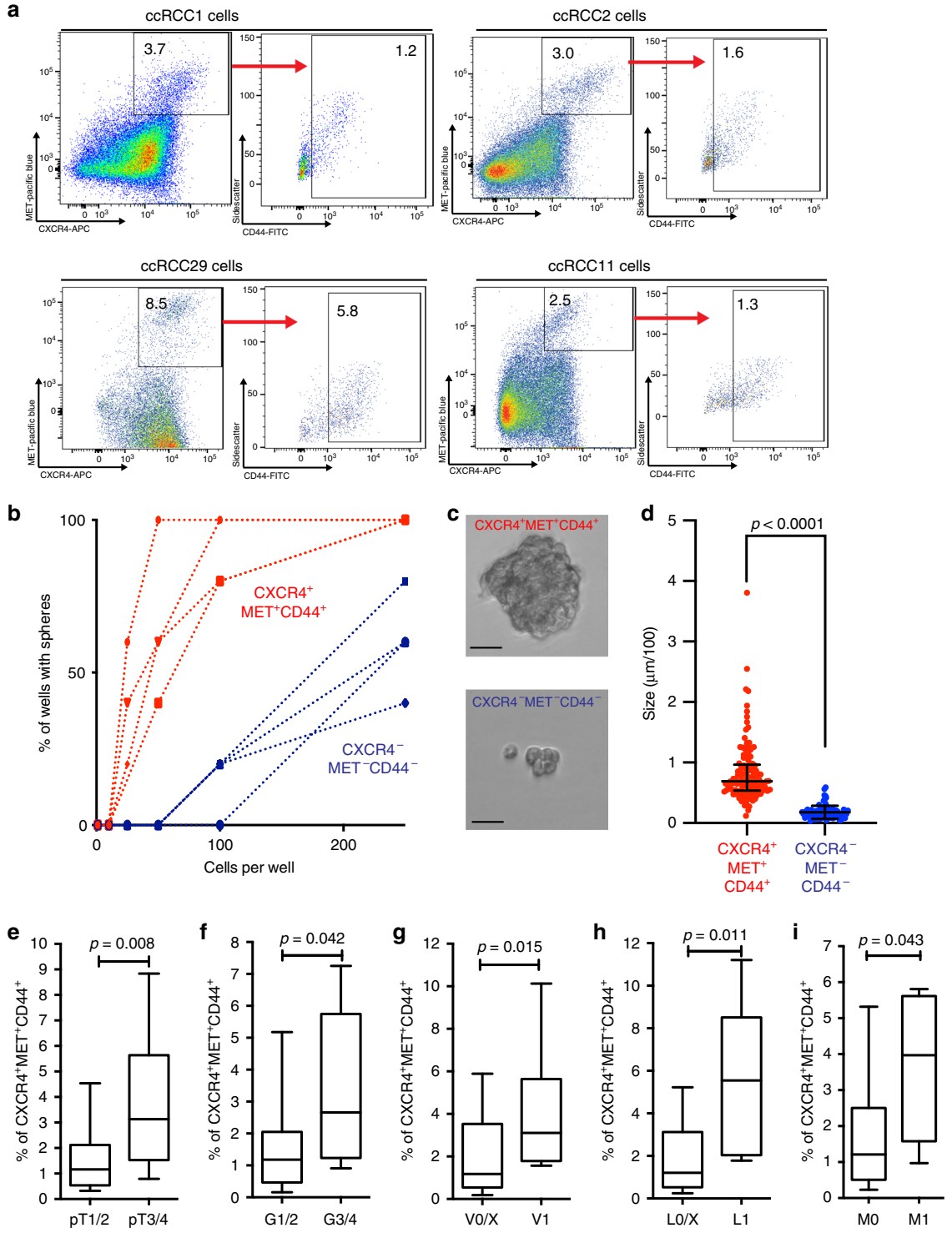

**Fig. 1 Analysis of ccRCC cancer stem cells. a** Representative FACS of single-cell suspensions of ccRCC cells using the markers CXCR4, MET, and CD44. Axis label of sidescatter is a multiple of 1000. **b** Limiting dilution assay of the sphere formation capacity of CXCR4$^+$MET$^+$CD44$^+$ and CXCR4$^-$MET$^-$CD44$^-$ cells ($n = 4$, single-cell suspensions of independent patients): 1–250 cells per well were seeded and cultured for 7 days, and all wells with >1 sphere were counted as positive. 10 wells per concentration were analyzed. **c** Representative image of spheres of the two phenotypes after 7 days of culture (Scale bar, 25 μm) ($n = 4$, single-cell suspensions of independent patients). **d** Quantification of sphere size of CXCR4$^+$MET$^+$CD44$^+$ and CXCR4$^-$MET$^-$CD44$^-$ cells after 7 days of culture ($n = 125$ CXCR4$^+$MET$^+$CD44$^+$ spheres and 83 CXCR4$^-$MET$^-$CD44$^-$ spheres). Shown is the diameter in μm/100, line represents median size and error bars show the interquartile range. $p$-value was calculated by two-sided Mann–Whitney test. Frequencies of CXCR4$^+$MET$^+$CD44$^+$ cells in ccRCCs from patients with (e) pathological stage (pT) 1/2 ($n = 22$) or 3/4 ($n = 19$), (**f**) Fuhrman grade (G) 1/2 ($n = 34$) or 3/4 ($n = 7$), (**g**) without ($n = 31$) or with ($n = 10$) venous invasion (V), (**h**) without ($n = 34$) and with ($n = 7$) lymph node invasion (L), and (**i**) without ($n = 33$) or with ($n = 8$) distant metastases (M) at the time of surgery. Numbers represent single-cell suspensions of individual patients. Boxes show the 25, 50, and 75 percentiles, whiskers the 10 and 90 percentiles. Outliers are shown as dots. $p$-values were calculated by two-sided Mann–Whitney test.

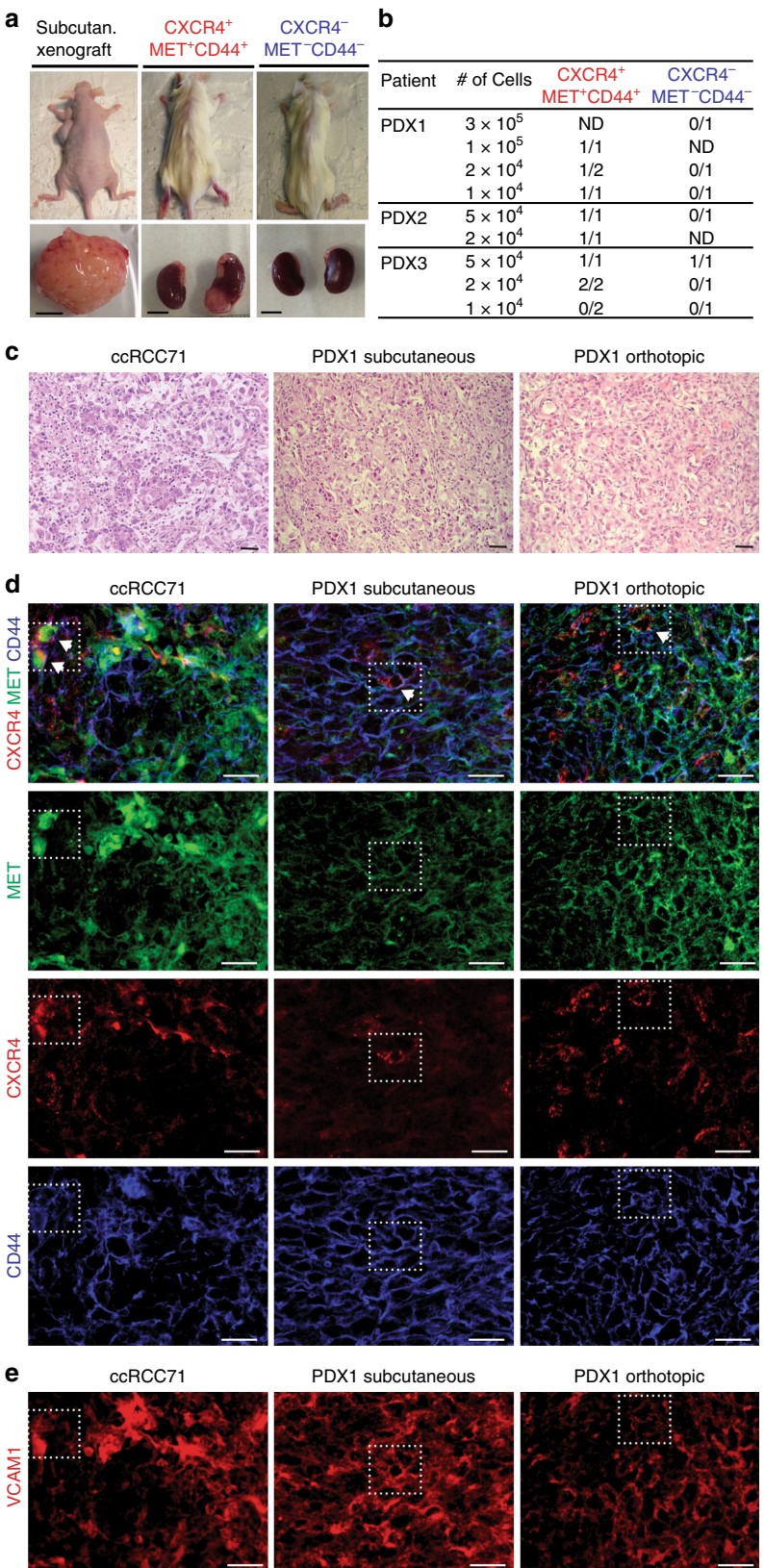

inter-patient differences in the expression of single markers. Few CD44−positive cells were detected in areas with a predominance of nested clear cells but were mostly present in dedifferentiated areas of the tumor (Supplementary Fig. 2d). MET was often strongly positive at tumor edges and more diffusely detected in the centers (Supplementary Fig. 2e). CXCR4 expression was detected in single cells, rather than clusters of cells, throughout the tumor (Fig. 2d).

VCAM1, which has been proposed to mark the cell-of-origin in ccRCC[31], generally overlapped with MET in primary tumors and xenografts (Fig. 2e). CXCR4+MET+CD44+ were mostly positive for VCAM1, suggesting that the latter might represent a

**Fig. 2 Xenotransplantation and in vivo localization of CXCR4+MET+CD44+. a** Representative images of xenotransplanted mice, and subcutaneous and orthotopic tumors (Scale bars, 0.5 cm). **b** Tumor formation by CXCR4+MET+CD44+ and CXCR4−MET−CD44− cells: cells were isolated from subcutaneous tumors, FAC-sorted and orthotopically transplanted into the renal parenchyma at the indicated cell numbers. Tumor formation was analyzed when mice showed symptoms or tumors were palpable. **c** Representative HE of primary ccRCC specimens and corresponding subcutaneous and orthotopic xenografts (scale bars, 50 μm). **d** Representative immunofluorescence of CXCR4, MET, and CD44. CXCR4+MET+CD44+ cells are highlighted in dotted lines and arrows. Immunofluorescence of each individual marker is shown for clarity (sale bars, 50 μm). **e** Representative immunofluorescence of VCAM1. Location of CXCR4+MET+CD44+ cells are highlighted in dotted lines (scale bars, 50 μm). Stainings were performed in three independent PDX and in 42 ccRCC specimens.

subpopulation of VCAM1+ cells in ccRCC. We further marked proximal tubules with lotus tetragonolobus lectin (LTL) and stained for the distal-tubule marker Calbindin. We detected LTL-positive cells in all tumors, albeit with varying frequency, but were unable to detect any Calbindin-positive tumor cells (Supplementary Fig. 2f, g). The specificity of both markers was confirmed in normal-adjacent tissue (Supplementary Fig. 2h, i). This confirms RNA sequencing data suggesting that ccRCC maintains the expression features of proximal tubule cells[31,32]. Together, these data show that high numbers of self-renewing CSCs correlate with the features of high tumor progression and metastasis.

**Characterization of ccRCC spheres and organoids.** We cultured CXCR4+MET+CD44+ cells as spheres under non-adherent conditions and as organoids in Matrigel (see scheme in Fig. 3a). Sphere and organoid culture conditions were similar to those previously described[11,33,34] but were adapted to the specific growth factor needs of kidney cancer cells. CXCR4+MET+CD44+ cells became enriched in sphere cultures (Fig. 3b and Supplementary Fig. 3a) and could be passaged without significant loss of spheres (Supplementary Fig. 3c). This demonstrates that sphere cultures enrich and maintain self-renewing CSCs. CXCR4+MET+CD44+ cells were also required for the initial formation of cultured organoids (Supplementary Fig. 3d), but their frequency quickly decreased, becoming comparable to those found in the primary tumors. This indicates that cells undergo differentiation in organoids (Fig. 3c and Supplementary Fig. 3b).

In sphere cultures, cells aggregated into solid structures (Fig. 3d), which contrasted with organoid cultures, where the majority of cells formed large hollow cysts (Fig. 3e). A subset of organoids exhibited other morphologies, such as more solid structures or intertwined tubes (Supplementary Fig. 3e). Such differences were observed within organoids of the same patient, as well as from different individuals. Immunofluorescence staining of spheres revealed weak E-cadherin (ECAD) with no preferential association to the cell surface (Fig. 3f, upper left). In contrast, in organoids E-cadherin located to lateral cell membranes (Fig. 3g, upper left) indicating epithelial cell differentiation. Carbonic anhydrase IX (CA9) staining confirmed that both types of cultures consisted of kidney cancer cells (Fig. 3f, g, upper right)[35]. LTL, which can be used to mark proximal tubule brush borders, only marked organoid cells (Fig. 3f, g, middle left). Although LTL staining was generally diffuse, it was localized apically in some of the organoids, which is typical for proximal tubules (Supplementary Fig. 3f). Most sphere cells were positive for CXCR4, MET, and CD44, but only a subset of cells of the organoids was positive for all three markers, further indicating differentiation of organoid cells (Fig. 3f, g, middle right and Supplementary Fig. 3h). VCAM1-positive cells were detected in spheres and organoids (Fig. 3f, g, lower left), and both sphere and organoid cultures contained Ki-67-positive cells (Fig. 3f, g, lower right).

Electron microscopy revealed that organoids contained an inner lumen (marked with L), and consisted of layers of tightly packed cells with apical brush borders (Fig. 3h), which confirms epithelial differentiation. Lateral contacts between cells consisted

of tight junctions (red arrowheads), adherens junctions (AJ) and desmosomes (D) (Fig. 3h). Cells on the basal side exhibited smooth basement membranes (Fig. 3h). Organoid cells contained glycogen deposits (asterisks) and lipid droplets (arrows), both typical characteristics of ccRCC[36,37]. The presence of glycogen was further confirmed by Periodic acid Schiff (PAS) staining (Supplementary Fig. 3g). In summary: markers, structures and junctions confirm that sphere cultures enrich for undifferentiated, self-renewing cells, while organoids show clear epithelial and kidney-specific cell differentiation.

**WNT and NOTCH are activated in kidney CSCs.** We carried out genome-wide expression profiling of FAC-sorted CXCR4+MET+CD44+ cells, sphere cells and non-sorted control cells from the tumors of three patients. A heatmap clearly distinguished sphere and CXCR4+MET+CD44+ cells from controls (Fig. 4a). Gene ontology (GO) analysis of the gene signature common to spheres and CXCR4+MET+CD44+ cells (all genes with FC > 1.5 or <−1.5 and p-value < 0.05 were considered) in either spheres and CXCR4+MET+CD44+ cells revealed an activation of genes associated with cell fate determination (i.e. KLF4, SOX9), kidney development (PAX2, SALL1), stem cell maintenance (PROM1, ALDH1A1) and chromatin modifications (MYST3) (Fig. 4b, left part and Supplementary Fig. 4a; see also Supplementary Data 1 for the full gene list). A Kegg pathway analysis[38] revealed further that the CSCs exhibited activation of WNT and NOTCH signaling, as seen by the top-scoring genes TCF7L2, TCF3, LGR4, AXIN2, and EP300 (WNT targets) and RBPJ, NOTCH3, HES1, and JAG1 (NOTCH targets) (Fig. 4b). Expression of these genes was validated in CXCR4+MET+CD44+ cells and sphere cultures derived from 10 further ccRCC tissues (Supplementary Fig. 4b, c). Luciferase reporter gene assays confirmed a significant activation of WNT and NOTCH components in spheres and CXCR4+MET+CD44+ cells, in contrast to adherent cells, which are not self-renewing[33,34] (Fig. 4c, d). Moreover, WNT and NOTCH target gene expression was lower in organoids (Supplementary Fig. 4d, e), again indicating that differentiated cells accumulate in organoids. A knockdown of CTNNB1 (β-catenin) and NOTCH1 by siRNA treatment strongly reduced the numbers of spheres in the cultures (Fig. 4e, f). We also examined the secretion of two WNT ligands, which showed high expression in the microarray in sphere and organoid culture supernatants using ELISA. The secretion of WNT10A but not WNT7B was elevated in sphere cultures when compared to the secretion in adherent or organoid cultures from the same patients (Fig. 4g, h).

In sphere cultures, we further observed that ectopic activation of WNT signaling using a β-catenin-LEF1 fusion protein[39] (Fig. 4i) triggered a strong induction of NOTCH signaling (Fig. 4j). In contrast, MAML, NICD, or NOTCH1 siRNA transfections had no significant effects on WNT signaling (Supplementary Fig. 4f–h). The WNT activation of NOTCH signaling was accompanied by transcriptional upregulation of JAG1 and HES1 (Fig. 4k). A lower WNT-dependent activation of NOTCH was observed in organoid cultures (Supplementary Fig. 4i–k). These data indicate that NOTCH is upregulated by

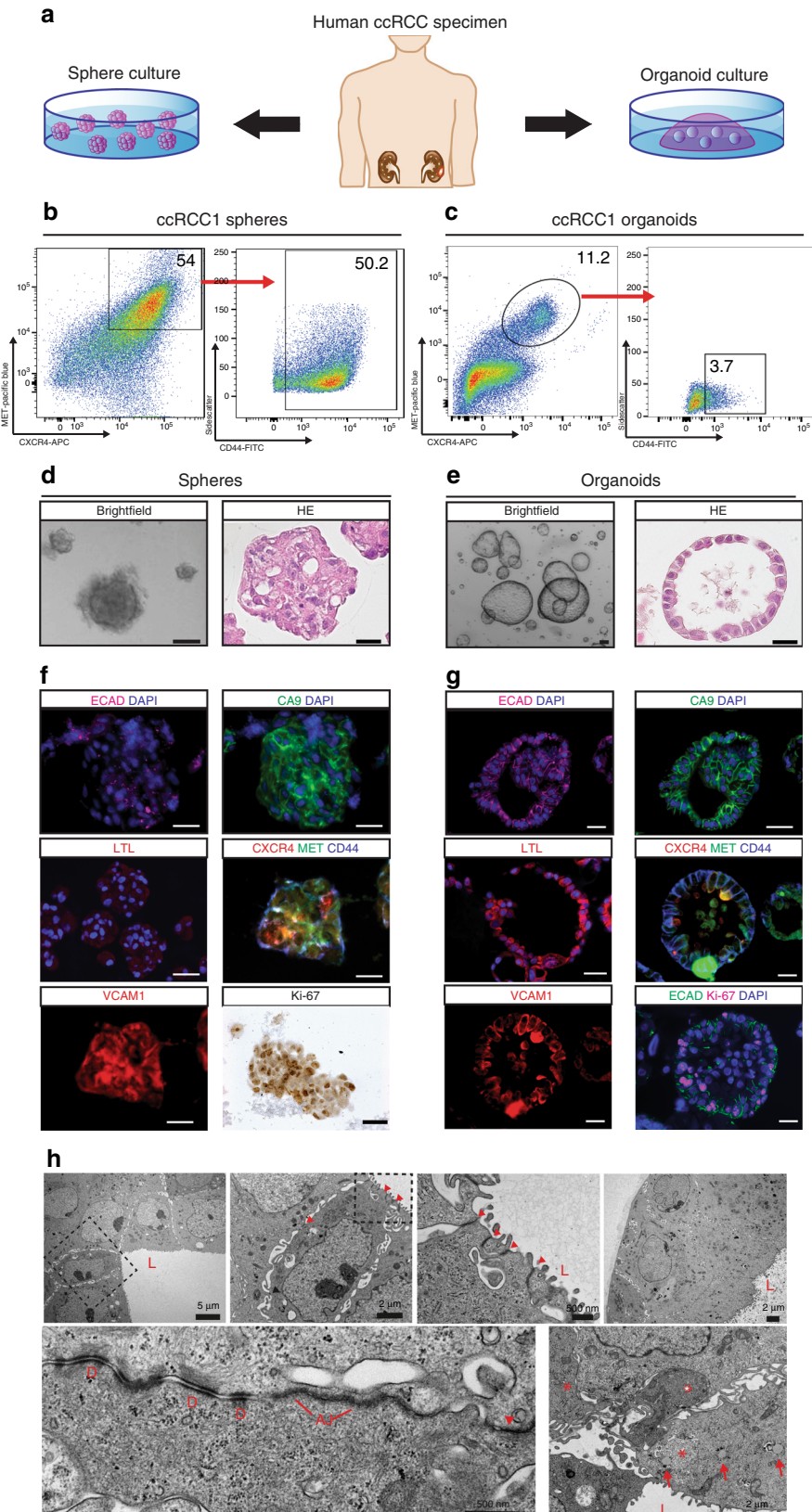

WNT signaling in renal cancer stem cells, mediated by the transcriptional upregulation of *JAG1*, a mechanism previously described in the intestine[40].

To confirm that the stem cell signature was associated with the clinical course of ccRCC, we examined gene expression data that had previously been obtained from 28 additional ccRCC patients (Supplementary Table 2)[41]. In these analyses, we included all genes with a FC > 1.5 or < −1.5 and *p*-value < 0.05. Supervised hierarchical clustering showed that tumors could be distinguished from normal tissues from the same patients (Supplementary

**Fig. 3 Characterization of ccRCC-derived sphere and organoid cultures. a** Human ccRCCs were collected at nephrectomy. Single cells were isolated and seeded as spheres in non-adherent conditions or as organoids in Matrigel. **b**, **c** Percent of CXCR4$^+$MET$^+$CD44$^+$ cells obtained by FACS after 7 days of sphere or organoid cultures. Axis label of sidescatter is a multiple of 1000. **d**, **e** Brightfield images and HE staining after 7 days of sphere and organoid culture (scale bars, 100 μm). **f**, **g** Immunofluorescence for Carbonic anhydrase IX (CA9), E-Cadherin (ECAD), LTL, CXCR4, MET, CD44, VCAM1, and Ki-67 (scale bars, 25 μM). Stainings were performed in five sphere and organoid cultures of five independent patients. **h** Transmission electron microscopy of representative organoid cultures (see scale bars for sizes): L, luminal side; D, desmosomes; AJ, adherens junctions; arrowheads, tight junctions; arrows, lipid droplets; asterisks, glycogen deposits. TEM was performed on organoid cultures of three selected patients.

Fig. 4l, top part, compare orange with blue). The heatmap revealed that the stem cell gene signature was differentially expressed in normal kidney tissues and tumors (Supplementary Fig. 4l). We selected all genes associated with kidney development, stem cell maintenance, WNT, and NOTCH signaling (all genes plotted in Fig. 4b) and compared the expression of these genes with patient survival. Kaplan–Meier survival analyses showed that the stem cell signature was associated with poor overall survival of patients (Supplementary Fig. 4m). Poor survival was also associated with the expression of individual WNT (DKK3) or NOTCH signaling (NOTCH3) genes (Supplementary Fig. 4n). We confirmed these data from our small patient cohort using RNA sequencing data from the TCGA KIRC study[3]. The stem cell signature (Fig. 4l), the combination of all deregulated WNT and NOTCH pathway genes (Fig. 4m), or DKK3 (Fig. 4n) and NOTCH3 (Fig. 4o) expression alone were associated with the overall survival of the 462 patients.

We next used the CEL-Seq technique[42,43] to carry out single-cell sequencing of CXCR4$^+$MET$^+$CD44$^+$ cells. The aim was to explore the extent of heterogeneity in ccRCC stem cell populations and to determine the fraction of the cells which exhibited WNT and NOTCH activation. We analyzed 90 cells per sample and combined both datasets to identify only those genes whose expression commonly varied in both datasets (6775 genes) (Supplementary Fig. 5a, b). Cluster analysis was performed and most significant canonical clusters were aligned and subjected to tSNE analysis[44]. This revealed three clusters (Fig. 5a, marked by different colors), which were not unique to individual patients, as cells from each sample were found in each of the clusters. We identified the top 20 genes of each cluster (Fig. 5b, indicated on the right). The genes included extracellular and transmembrane genes and extracellular vesicles, such as CXCL1, CXCL6, MMP7, CD70, and COL5A1. While the cancer stem cell genes and WNT and NOTCH signaling genes were not included in this list, we found that clusters 1 and 3 showed higher expression of WNT and NOTCH pathway genes, stem cell genes and kidney-specific genes, for instance LGR4, TCF7L2, JAG1, ALDH1A1, and PAX2 (Fig. 5c). Other stem cell-specific genes were expressed across all three clusters (Supplementary Fig. 5c). These results indicate that while cluster 1 and 3 have the highest expression of certain CSC genes, in general all clusters share expression of a subset of CSC genes. Interestingly, WNT and NOTCH genes are both activated in the same cell clusters and do not mark different subpopulations of cells.

Collectively, we conclude that spheres and CXCR4$^+$MET$^+$CD44$^+$ cells share a gene signature that distinguishes them from controls and characterizes stem cells in renal carcinoma. When we compared this with patient data, we observed that the activation of the stem cell signature as well as individual genes involved in WNT and NOTCH signaling were associated with worse survival. The connection between WNT/NOTCH signaling and the phenotype suggested that these pathways could represent a weakness that might be exploited in targeting CSCs and PDXs.

**Inhibitors of WNT and NOTCH reduce CSC and organoid growth.** We examined the effects of inhibitors of WNT and

NOTCH signaling in all our experimental models: sphere and organoid cultures and PDXs. First, sphere cultures were treated with the WNT inhibitor ICG-001[45] or the NOTCH inhibitor DAPT[46]. We analyzed sphere cultures of a subset of 41 patients from the total of 55 (Supplementary Table 1). Cultures from a majority of patients responded to ICG-001, while DAPT inhibited cultures from three other patients, both in concentration-dependent manners and within one week (Fig. 6a, b). A few cultures failed to respond (Supplementary Fig. 6a, d). Sphere cultures with ICG-001 IC50 < 20 μM or DAPT IC50 < 30 μM were considered as responders. The responses could be classified into four subgroups: ICG-001 responders (71%), DAPT responders (63%), ICG-001/DAPT double responders (46%) or non-responders (12%) (Fig. 6c, Supplementary Table 3). Responses to either ICG-001 or DAPT did not correlate with the pathological stage or grade of the cancer or percentage of CXCR4$^+$MET$^+$CD44$^+$ cells (Supplementary Table 4) indicating that specific mechanisms of response exist. We also applied combinations of the two inhibitors at low concentrations for which single inhibitors produced only minimal effects (Fig. 6d). Combinations had a strong effect on ICG-001/DAPT double responders, but were less effective in non-responders indicating that double responders might also benefit from the combined treatment with both inhibitors (Fig. 6d). ICG-001 suppressed WNT target genes and DAPT suppressed NOTCH target genes in responding sphere cultures (Fig. 6e, f), but not in non-responders (Supplementary Fig. 6b, d). We also examined sphere cultures using other inhibitors directed against the receptors used above for sorting CSC, but these were less effective than WNT and NOTCH inhibition: the MET inhibitor Crizotinib[47] inhibited sphere growth in 44% of specimens, and the CXCR4 chemokine receptor inhibitor AMD3100[48] in 17%.

We further analyzed the effects of inhibiting the two pathways in ccRCC organoid cultures: WNT inhibitors had potent effects on the organoids, but the response to NOTCH inhibitors was weak (Fig. 6g, h). Moreover, ICG-001 suppressed WNT target genes in organoids, while DAPT had little or no effect (Fig. 6i, j). We reasoned that WNT inhibition had a strong effect on both self-renewal and differentiation, which is supported by the fact that ICG-001 inhibited growth in both spheres and organoids. NOTCH inhibition, in contrast, could have one of two effects: either it targeted more specifically self-renewing cells and had a weaker effect on their differentiation during organoid formation, or it had more moderate effects on these cells, permitting their differentiation before they were targeted by the treatment. We therefore reseeded DAPT-treated organoids and observed an impaired secondary growth of organoids (Supplementary Fig. 6e). We conclude that in the initial passage, DAPT treatment may have reduced the CSC pool preventing cells to regrow in the second passage. To exclude non-specific toxicity, we treated spheres and organoids derived from normal-adjacent tissue. DAPT had no significant effect on the five sphere and organoid cultures that were examined (Supplementary Fig. 6f). ICG-001 also produced no significant toxicity in sphere cultures, but led to a reduction of organoid growth (Supplementary Fig. 6g).

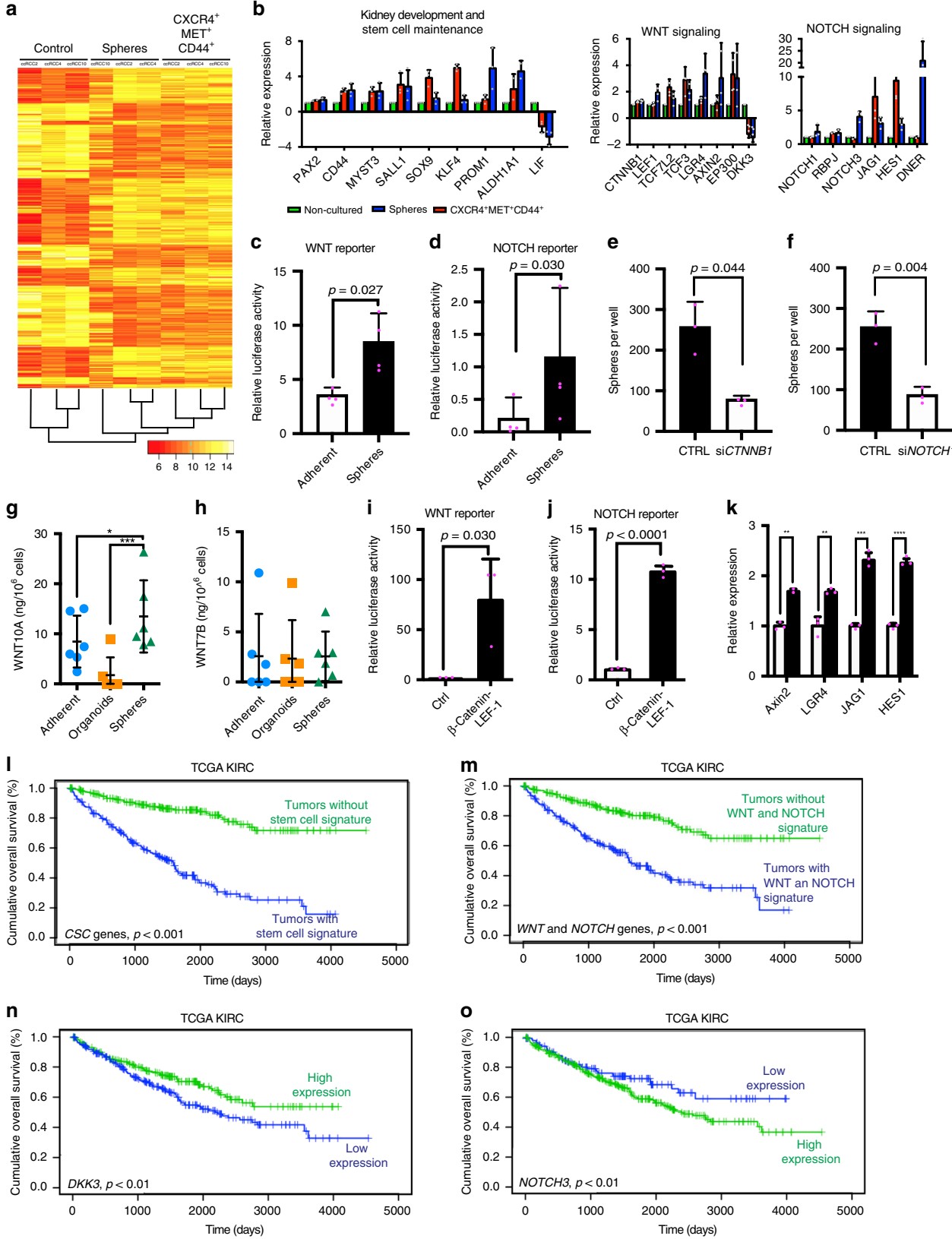

To exclude that the effects of ICG-001 were due to unspecific events, we tested additional WNT inhibitors in specimens that responded to ICG-001: the TANKYRASE inhibitor XAV939[49], the β-CATENIN inhibitor LF3[50], and the Porcupine inhibitor C59[51]. While each of the WNT inhibitors reduced sphere formation and organoid growth, they showed different efficiencies, decreased the expression of WNT target genes to varying levels, and some had stronger effects on normal cells. This indicates that ICG-001 is the most suitable inhibitor for further experiments (Supplementary Fig. 6h–j). The MAML inhibitor IMR-1 showed strong effects both on spheres and organoids. In general, inhibition by IMR-1 was stronger than

**Fig. 4 Activation of WNT in NOTCH signaling in CXCR4+MET+CD44+ and spheres.** Microarray analysis of spheres, FAC-sorted CXCR4+MET+CD44+ and control cells from three ccRCCs. **a** Heatmap and hierarchical clustering of the 500 most variable probes. **b** log2 fold changes of top-scoring genes associated with the GO terms kidney development, stem cell maintenance, WNT and NOTCH signaling. Data are shown as mean, error bars represent s.d. ($n = 3$ patients). **c, d** WNT or NOTCH luciferase reporter gene assays of sphere and adherent cultures of ccRCC ($n = 5$ independent patients). Data are shown as mean, error bars represent s.d., $p$-values calculated by two-sided $t$-test. **e, f** siRNA knockdown of CTNNB1 or NOTCH1 in spheres ($n = 5$ patients). Data are shown as mean, error bars represent s.d., $p$-values calculated by two-sided $t$-test. **g, h** ELISA for WNT10A and WNT7B secretion of adherent, organoid and sphere cultures ($n = 6$ patients). Supernatants were collected after 7 days of culture. Values were normalized to the number of cells per ml medium and are represented as ng protein per $10^6$ cells. Line represents mean protein concentration and error bars show the s.d. $p$-values: *<0.05, ***<0.001 by RM ANOVA with Dunnett's post-test (two-sided). **i, j** WNT or NOTCH luciferase reporter gene assays of sphere cultures ($n = 5$ independent patients) transfected with a β-Catenin-LEF1 fusion protein encoding plasmid. Data are shown as mean, error bars represent s.d., $p$-values calculated by two-sided $t$-test. **k** Expression of WNT and NOTCH pathway genes measured by RT-qPCR in β-Catenin-LEF1-transfected spheres ($n = 5$ patients). Data are shown as mean, error bars represent s.d., $p$-values: **<0.01, ***<0.001, ****<0.0001 by two-sided, paired $t$-test. Overall survival of TCGA KIRC specimens with or without the stem cell signature (**l**), the WNT and NOTCH signature (**m**), or high (>median) or low (<median) DKK3 (**n**) or NOTCH3 expression (**o**). Stem cell signature predictor score was calculated by multivariate Cox regression, patients were stratified into groups according to the median predictor score. Censored patients are presented as vertical lines. Significance was tested by log rank test (two-sided).

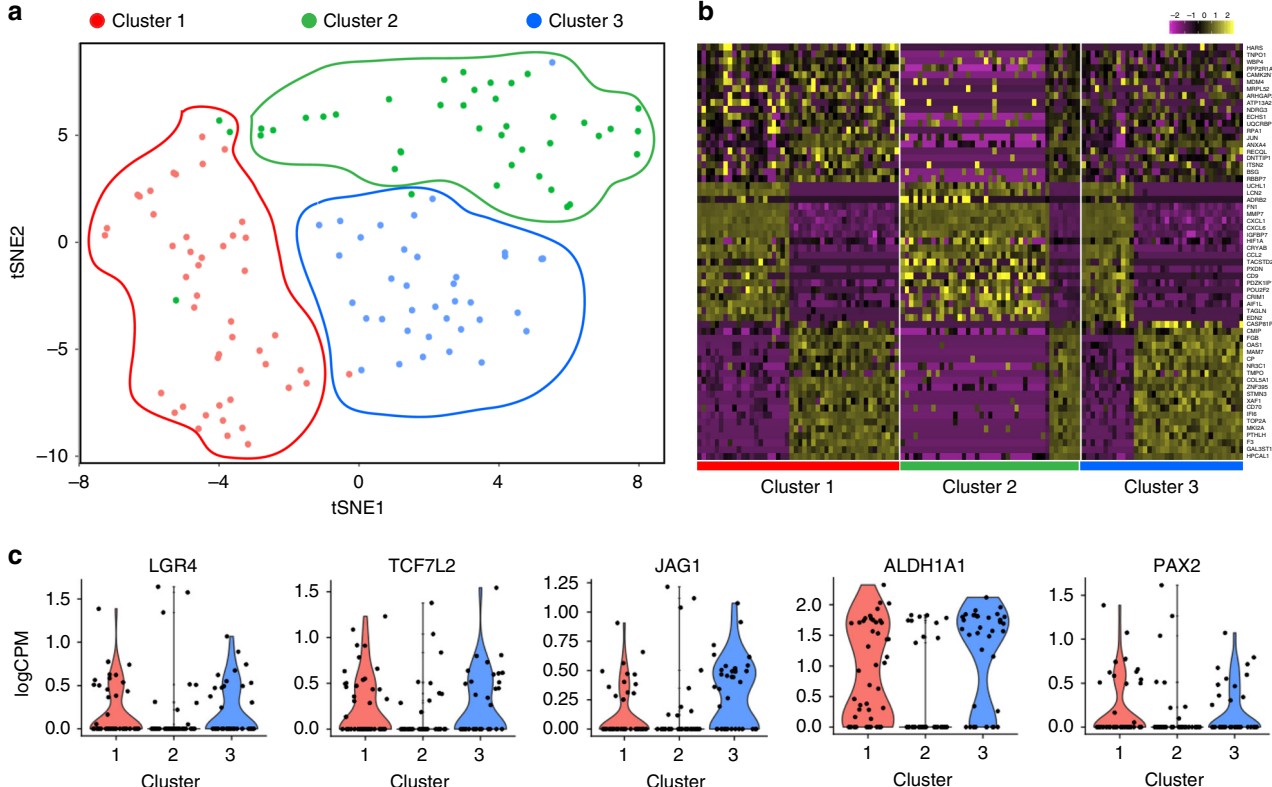

**Fig. 5 Single-cell sequencing of CXCR4+MET+CD44+ cells.** Single-cell RNA-Seq was performed on 90 cells per patient by using the CEL-Seq2 method. Canonical cluster analysis was performed on common variable genes in the combined datasets, and tSNE clusters were subsequently identified. **a** tSNE plot of single cells, clusters marked in different colors, **b** Heatmap of the top 20 markers for each cluster. **c** Expression of WNT, NOTCH and stem cell marker genes in each cluster. Data are shown as scatter plot, violin plot represents data density.

the effects observed by DAPT, specifically in organoids, yet it also had pronounced effects on normal cells (Supplementary Fig. 6k).

**Inhibitors of WNT and NOTCH signaling reduce PDX growth.** We also examined the effects of WNT and NOTCH inhibition in our third model, PDXs. PDX (PDX1 and PDX4) were re-passaged, and once tumors became palpable, animals were allocated to different systemic treatments. PDX4 mice showed a marked reduction of tumor volumes when treated with ICG-001 (Fig. 7a and Supplementary Fig. 7a); at 26 days, their tumors were 44% smaller than those of control animals. The NOTCH inhibitor DAPT had no significant impact on

the tumor volumes (Fig. 7a). We examined the tumors histologically following treatment: we found that large areas of the tissue became fibrotic upon ICG-001 treatment (Fig. 7c, middle row). At the cellular level, these tumors contained fewer proliferating cells, as determined by staining with Ki-67 (Fig. 7c, middle row).

Next, PDX1 animals were subjected to single ICG-001 and DAPT and combined therapies. Treatment with ICG-001 led to strong reductions of tumor volumes; DAPT had less pronounced effects (Fig. 7b and Supplementary Fig. 7b). At day 93, ICG-001 and combined treatments reduced tumor volumes by 90%, while DAPT reduced them by 52% (Fig. 7b and Supplementary Fig. 7b). Histological examinations revealed that ICG-001- and DAPT-treated tumors contained necrotic regions and a

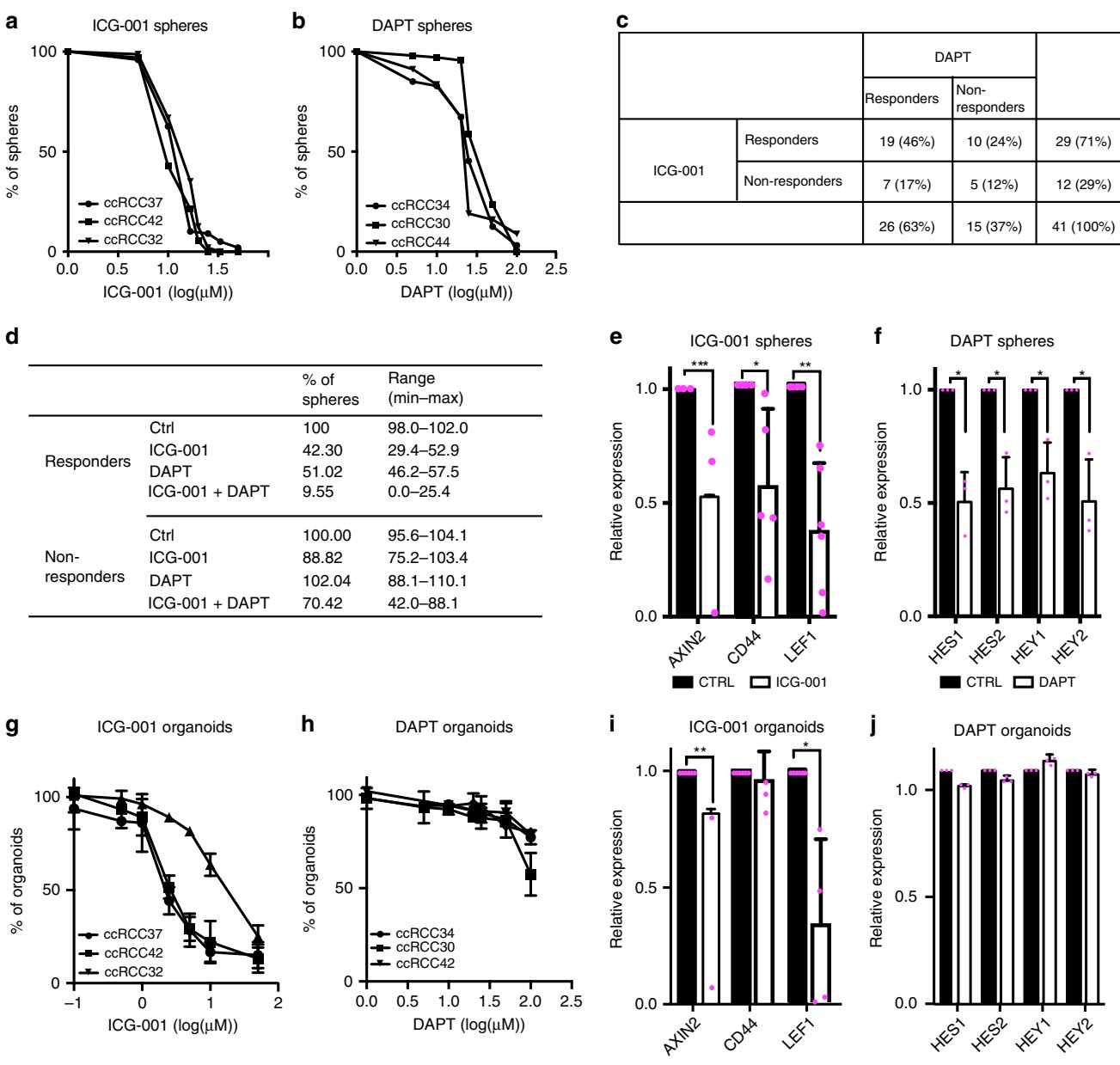

**Fig. 6 Pharmacological inhibition of WNT and NOTCH signaling in ccRCC spheres and organoids. a**, **b** ccRCC sphere cultures were treated with 5–50 µM ICG-001 or 1–100 µM DAPT for 7 days, and spheres <25 µm were then counted. Data are shown for three exemplary cultures. Experiments were performed in 41 ccRCC sphere cultures. **c** Overview of DAPT and ICG-001 responders in 41 ccRCC spheres. Sphere cultures with ICG-001 IC50 < 20 µM or DAPT IC50 < 30 µM were classified as responders. **d** ICG-001 and DAPT responders or non-responders were treated with 20 µM ICG-001, 20 µM DAPT or both ($n = 5$ for each treatment). Sphere numbers were counted after 7 days. **e**, **f** Sphere cultures treated with 20 µM ICG-001 or 20 µM DAPT ($n = 4$ for each group), and the expression of target genes was measured by RT-qPCR. Data are shown as mean, error bars represent s.d. p-values: *<0.05, **<0.01, ***<0.001 by two-sided t-test. **g**, **h** ccRCC organoid cultures were treated with 0.1–50 µM ICG-001 or 1–100 µM DAPT for 7 days. Metabolic activity was measured by CellTiterGlo assay and normalized to vehicle-treated controls. Data are shown for three exemplary cultures as mean ± SD of three technical replicates. Experiments were performed in 15 organoid cultures. **i**, **j** Organoid cultures were treated with 5 µM ICG-001 or 25 µM DAPT, and the expression of target genes was measured by RT-qPCR ($n = 3$ independent patients). Data are shown as mean, error bars represent s.d., p-values: *<0.05, **<0.01 by two-sided t-test.

remarkable reduction in numbers of proliferating cells (Fig. 7d, middle and lower rows).

Immunostaining for β-CATENIN and in situ hybridization for AXIN2 in PDX4 showed that after ICG-001 treatment, tumor tissues displayed a reduced expression of these markers, in contrast to vehicle-treated controls (Fig. 7e). In contrast, DAPT treatments did not markedly alter NOTCH1 and JAG1 expression; in fact, the nuclear localization of NOTCH1 was

even more pronounced (Fig. 7g). Similar staining procedures in PDX1 confirmed a reduction of all four markers under treatment with either ICG-001 or DAPT, corresponding to the effects on tumor volumes (Fig. 7f, h). FACS experiments in PDX4 confirmed that effective WNT inhibition was accompanied by reduced numbers of CXCR4[+]MET[+]CD44[+] cells, while DAPT treatment resulted in higher numbers of these cells (Supplementary Fig. 7c). A comparative expression analysis of

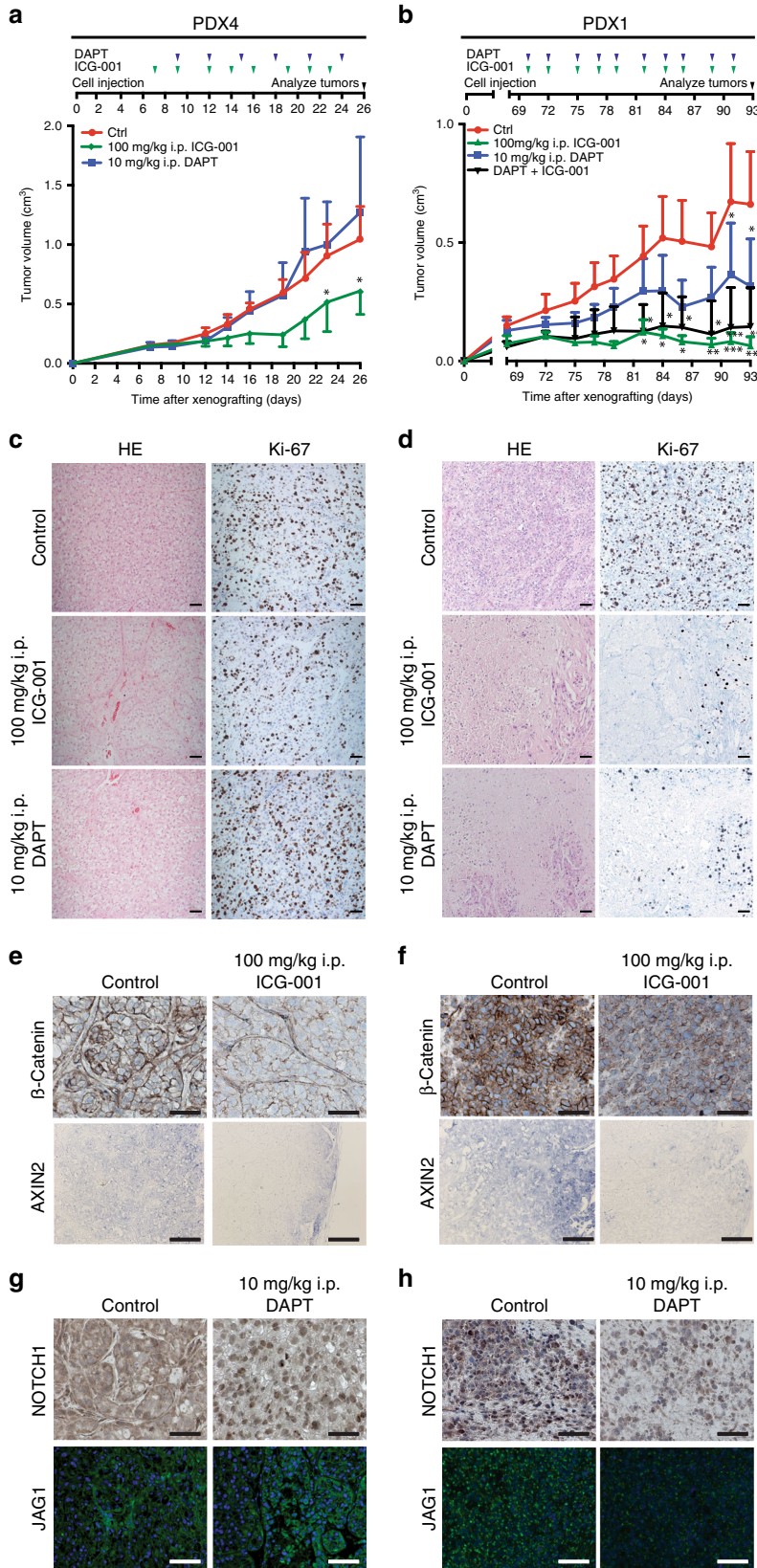

WNT and NOTCH genes showed a markedly lower expression of *JAG1* in PDX4, which potentially explains the lower response to DAPT inhibition (Supplementary Fig. 7d). The responses of the two types of xenografts were similar to those of sphere cultures from corresponding patients (Supplementary Fig. 7e),

confirming that sphere cultures are useful surrogates for tumor responses.

PDX1 tumors were also subjected to treatment with the clinically approved substances sunitinib, sorafenib, bevacizumab, and everolimus. Responses were assessed according to the

**Fig. 7 Inhibition of WNT and NOTCH signaling in patient-derived ccRCC xenografts. a, b** Treatment schemes and quantifications of tumor volumes of PDX4 and PDX1 tumors ($n = 3$ tumors in each group) treated with 100 mg/kg ICG-001 (green diamond), 10 mg/kg DAPT (blue squares), combination (black diamond, PDX1 only) or vehicle (red circles) every three days. Data are shown as mean, error bars represent s.d., $p$-values: *<0.05, *<0.01, ***<0.001 by two-way ANOVA with Tukey post-test (two-sided). **c, d** HE and Ki-67 staining of representative sections of tumors of PDX1 and PDX4 (scale bars, 50 μm). **e, f** Immunofluorescence of β-CATENIN (scale bars, 50 μm) and in situ hybridization for AXIN2 (scale bars 100 μm) in PDX4 and PDX1 tumors in mice treated with 100 mg/kg ICG-001. **g, h** Immunofluorescence of NOTCH1 and JAG1 in PDX4 and PDX1 tumors in mice treated with 10 mg/kg DAPT (scale bars, 50 μm).

RECIST criteria[52] as progressive diseases (personal communication with EPO). We did not observe any signs of unspecific toxicity in mice at the end of the treatment period, as assessed by weight loss, general appearance, and macroscopic examination of visceral organs. Both inhibitors have also been used in other preclinical models in the same or lower concentrations[53,54]. In conclusion, xenograft treatments provide further evidence that specific patient groups could profit from therapeutic WNT and NOTCH inhibition.

## Discussion

CSCs play a central role in tumors' resistance to therapies and their metastatic potential, and treatments are unlikely to become cures unless they target this subpopulation. Work on human ccRCC, however, had not led to a clear characterization of CSCs[8,9,22,55] or to concepts to targeting them therapeutically. We identified a subpopulation of self-renewing CSCs that could be identified by a CXCR4+MET+CD44+ profile. Analyses of gene and protein expression revealed an activation of WNT and NOTCH signaling in these CSCs. Inhibition of WNT and NOTCH signaling through small molecular weight compounds blocked CSCs.

Current treatments for human ccRCC include various combinations of antiangiogenic drugs, mTOR inhibitors, and immunotherapy drugs that extend the lives of many patients[5]. However, these treatments rarely lead to complete remission, and nearly all of the tumors progress within 2 years[7]. Our work indicates that WNT and NOTCH signaling exert essential functions in CSCs and thus represent a weakness that could be therapeutically exploited.

We developed treatment regimens based on small molecule inhibitors that target components of WNT and NOTCH signaling. We applied the inhibitors to three model systems developed from tumors of ccRCC patients. CSCs produced nonadherent, self-renewing sphere cultures, and organoids, which differentiate to recapitulate the heterogeneous cell types observed in patients' primary tumors. The third method was to develop PDX animals by transplanting aggressive ccRCCs into immune-compromised mice.

Two other methods to culture ccRCC organoids have recently been described[56,57]. The first yielded organoids with a limited expansion capacity and a low rate of success. Those organoids showed less epithelial polarization than the organoids we produced here. Interestingly both previous methods described a maintenance of genomic clones in the organoids, supporting the assumption that organoids are representative of the primary tumors. Future work will be needed to clarify the differences between the cultivation methods and their significance.

Each of our models diverges from human disease in ways that could not be predicted without direct comparisons to each other and findings from patients. A crucial factor in the development of personalized approaches to the treatment of ccRCC will be the speed at which models can be developed from individual patients' diseases. The sphere and organoid cultures require one week for isolation and expansion and one week for attempts at treatment.

We regard this as an important step toward the translation of these methods into early phase clinical trials.

We could distinguish four classes of sphere cultures based on the type of inhibition to which they responded: WNT responders (71%), NOTCH responders (63%), WNT/NOTCH double responders (46%) or non-responders (12%). Organoids were highly sensitive to treatment with WNT inhibitors, while treatment with the NOTCH inhibitor DAPT had a markedly weaker effect. This may mean that WNT is required for both self-renewal and differentiation, while NOTCH is mainly involved in self-renewal. This notion is supported by our observation that pre-DAPT-treated organoids cannot be re-seeded, arguing that they undergo a depletion of the stem cell pool. Kidney PDX models are difficult to generate, but the tumors we could examine responded to both WNT and NOTCH inhibition in a manner comparable to the responses of sphere cultures.

Beyond the WNT and NOTCH responders, we identified smaller subgroups of CSCs that were inhibited by targeting other CSC-relevant molecules: CXCR4 with AMD3100 and MET with Crizotinib. While the results suggest that targeting surface markers of CSCs is less effective than targeting their downstream pathways, it also suggests that specific subgroups of patients might benefit from these strategies. Our work thus suggests that defining the features of patients' CSCs may make it possible to rapidly stratify their tumors as a route to better treatments[5]. It should also help distinguish patients likely to benefit from therapy from those who should be spared[58–60]. The mechanisms by which these tumors develop resistance to therapies will need to be further explored.

We found higher numbers of CSCs in cases of ccRCC characterized by high pathological stages and metastases. Their presence is predictive providing the CSCs have been accurately defined. CD105 and CXCR4 alone, used in previous studies to identify CSCs in kidney cancers[8,9], were less predictive of tumor progression. Kaplan–Meier survival analyses revealed an association between our new stem cell signature and patients' survival. This confirms a trend observed in other tumors, such as breast cancer, where stem cell signatures correlated with tumor aggressivness[61].

A major limitation of this study is the low number of xenografts used to test tumor-initiating capacity. In our hands, xenografts of RCC grew slowly with 3-month latency until the formation of subcutaneous tumors, which limited expansion of these tumors. After sorting for CSCs, cell numbers were limited and did not allow for three technical replicates per concentration.

We also used single-cell sequencing to determine the degree of heterogeneity within the CSC population. This yielded three subpopulations, two of which exhibited high expression of markers known to be associated with stem cells and kidney development. These two populations also showed a high expression of WNT and NOTCH signaling components and high expression from the corresponding target genes.

Our data help establishing a crucial connection between WNT and NOTCH signaling and the biology of CSCs in ccRCC. In contrast to *VHL*, *WNT*, and *NOTCH* mutations are rare in ccRCC[2,3]. We propose that cancer stem cells use autocrine

mechanisms to upregulate WNT signaling, as shown by the higher secretion of WNT10A and by the fact that NOTCH signaling is activated downstream of WNT. WNT10A has previously been suggested as an oncogenic WNT ligand in ccRCC[62], but it had not been linked with CSCs. Several other mechanisms involving WNT and NOTCH upregulation have been described in kidney cancer. VHL and other crucial drivers of kidney cancer activate both the WNT and NOTCH pathways. For instance, the loss of VHL stabilizes β-CATENIN via JADE1[63]. Activated β-CATENIN has been associated with advanced kidney cancers and lower patient survival[64]. WNT signaling can also be promoted by hyper-activated MET observed in ccRCC[65]. NOTCH signaling is activated through VHL/hypoxia signaling[66] or independently of VHL[67]. These findings are consistent with our results and the effects of inhibiting these pathways in our model systems. We interpret this to mean that they support new strategies to treat ccRCC based on the disruption of crucial mechanisms in cancer stem cells that play an essential role in ccRCC.

## Methods

**Patient samples**. ccRCC specimens were collected at the Department of Urology, Charité-University Medicine, Berlin (characteristics in Supplementary Table 1). The project was approved by the ethics committee of the Charite—University Hospital (EA1/134/12) and informed consent was obtained from all patients. Samples were processed within 24 h after surgery. The tissue was extensively washed in PBS, minced into cubes smaller than 1 mm³, digested enzymatically using collagenase P and filtered through sieves with 100 and 40 μm pore size. Subsequently, erythrocytes were lysed and leukocytes were depleted by MACS using anti-CD45 micro-beads (Milteniy BioTec). The selection of samples for specific analysis was based on tissue availability.

**Primary culture**. Tumor cell suspensions at 100.000 cells/ml were seeded as non-adherent sphere cultures in Poly-HEMA-pretreated (Sigma Aldrich) 24-well plates (Fisher Scientific) in DMEM/F12, supplemented with 20 ng/ml EGF, 20 ng/μl FGFb, 4 μg/μl Heparin, 1× B27 (Fisher Scientific), 1× Penicillin/Streptomycin (Gibco), and 125 μg/ml Amphotericin B (PAA). Cells were cultured for 7 days before harvesting or re-passaging.

For organoid cultivation, fresh tumor cell suspensions were prepared as described above and seeded at 15.000 cells per well in 25 μl growth factor-reduced Matrigel (Corning) drops (75% Matrigel, 25% growth medium) in 48-well plates. After polymerization, Matrigel lenses were overlaid with 250 μl DMEM/F12 supplemented with 20 ng/ml EGF, 20 ng/μl FGF, 4 μg/μl Heparin, 1× B27 (Fisher Scientific), 1× Penicillin/Streptomycin (Gibco), and 1.25 μg/ml Amphotericin B (PAA). Medium was changed every other day. Passaging of organoids was performed every 7 days: Matrigel was broken up by pipetting up and down several times, and organoids were collected in a tube. After centrifugation at 300g for 5 min, organoids were dissociated in TrypLE (Fisher Scientific) for 10 min. Cell cluster were reseeded as described above and a typical split ratio of 1:3. Organoids were filtered using a 40 μm mesh in the first passage to clear the culture from remaining single cells. Several downstream experiments demanded seeding of single-cell suspension. In this case, organoids were dissociated in TrypLE for 30 min. Early passage organoids were frozen in Recovery Cell Culture Freezing Medium (Thermo Fisher) according to the manufacturer's recommendations. A detailed protocol will be provided by the authors upon request.

Adherent cultures were grown in MEM in standard tissue culture plates (Fisher Scientific) with Earle's salts supplemented with 10% FCS, 1× NEAA, 2 mM L-Glutamin, 1× Penicillin/Streptomycin and 1.25 μg/ml Amphotericin B. Medium was changed every 3 days and cultures were passaged at 80–90% confluency.

**Fluorescence-activated cell sorting**. Isolated tumors cells were analyzed with FACS Aria I or Aria F Cell Sorters (BD Biosciences, Germany). Single staining was performed for each marker for correct instrument setup. Stem cells were sorted by anti-CD44-FITC (BD Biosciences), anti-CXCR4 (R&D Systems), biotin-anti-mouse (Sigma), Streptavidin-APC (Life Technologies), and anti-MET antibodies (Santa Cruz), anti-rabbit-Pacific Blue. Viable cells were identified by 7AAD staining (BD Biosciences). Single cells were gated by plotting SSC versus FSC and confirmed by gating FSC-A vs FSC-H and SSC-A vs. SSC-H. Viable cells were identified by negative 7AAD staining. Gating strategy is shown for one patient in Supplementary Fig. 8. Sorting efficiency was confirmed by resorting a subset of sorted cells and was >95% for all experiments. For the remaining markers single stainings were performed (see Supplementary Table 6 for a full antibody list).

**Gene expression analysis**. RNA was isolated from uncultured and unsorted control cells, CXCR4⁺MET⁺CD44⁺ or sphere-cultured cells using the RNeasy Mini Kit (Qiagen) and hybridized to the HumanHT-12 v4 bead chip (Illumina

Inc.) according to the manufacturers protocol. HumanHT-12 v4 bead chip data were normalized and log-transformed in Partek Genomics Suite (Partek) and analyzed using the limma package[68] in R. We identified a stem cell signature that included all genes that were upregulated in either spheres and CXCR4⁺MET⁺CD44⁺ with FC > 1.5 or <−1.5 and p-value < 0.05. GO term enrichment and Kegg pathway analyses were performed with DAVID[38]. The gene expression analyses of 28 fresh frozen ccRCC specimens has been performed previously at the Charité-University hospital and only the data were used for this study (GEO Accession Code: GSE66270 and GSE66271, for patient information see Supplementary Table 2)[41]. Expression of stem cell genes was analyzed using the GeneSpring GX software (Agilent), and supervised hierarchical clustering was performed in Genesis[69].

**Single-cell sequencing**. Single-cell analysis was performed using the CEL-Seq2 method published previously[42]. Single CXCR4⁺MET⁺CD44⁺ cells were sorted into 96-well plates containing all reagents for reverse transcription. Primers with unique cell barcodes for reverse transcription can be found in the publications[42,43]. SuperScript II (Thermo Fisher) was used for reverse transcription. AmPure XP beads (Beckman Coulter) was used for RNA clean-up. Single-cell RNA amplification was performed by in vitro transcription using the MEGAscript T7 transcription kit (Thermo Fisher). Amplified products were further proceeded to library preparation using Illumina sequencing primers. Paired-end sequencing was performed on a HiSeq 2500 sequencing system in a manner of 16 bases for read 1 (barcode read), 7 bases for the Illumina index and 51 bases for read 2.

Single sequencing raw BCL files were converted to FASTQ using the BCL2FASTQ software (Illumina). FASTQ files were processed using a multistep python library available on GitHub (https://github.com/yanailab/celseq2). Briefly, data were de-multiplexed using the barcode from read 1, and UMIs were attached to read 2 metadata. Using BowTie2, reads were mapped to the Hg19 reference genome, and raw reads were counted using a modified HTSeq-count script. Data were then analyzed using the Seurat package in R[70] (http://satijalab.org/seurat/). All cells expressing less than 200 genes and all genes expressed in less than three cells were excluded from further analysis. The datasets of both samples were combined for all downstream analysis. All data were scaled so that each gene had a mean expression of 0 and a variation of 1 across all cells. Reads were log-transformed and normalized to the total reads per sample to obtain RPM. To find clusters across samples, we performed canonical correlation analysis and forwarded these clusters to tSNE analysis. Cells whose variance could not be explained with greater 2-fold variance in comparison to PCA were excluded from the tSNE plot.

**Real-time quantitative polymerase chain reaction**. Total RNA from sphere cultures, organoids, and adherent cells was isolated using TRIZOL (Invitrogen), and 1 μg total RNA was reverse transcribed using MMLV reverse transcriptase according to the manufacturer's instructions (Promega). RT-qPCR was performed using the iCycler IQTM 5 multicolor real-time detection system (Bio-Rad), with absolute SYBR green fluorescein (ABgene) using a standard protocol. Primer sequences are listed in Supplementary Table 5. Primer specificity was tested by melting curve analyses and gel electrophoreses. Expression values were normalized to the endogenous control GAPDH and amplification efficiency using an adjusted $2^{-\Delta\Delta Ct}$ method.

**siRNA transfection and luciferase reporter gene assay**. Adherent cultures were transfected with 30 pmol siRNA oligos against CTNNB1, NOTCH1, or a control oligo by using Lipofectamine 2000 according to the manufacturer's instructions (Invitrogen). Twenty four hours after transfection 100,000 cells/ml were seeded as spheres in 24-well plates or reseeded in 6-well plates under adherent conditions. Five days after transfection, spheres were counted under a microscope. Three technical replicates per primary cell culture and group were analyzed. Knockdown efficiency was analyzed by Western blotting and RT-qPCR.

Adherent cultures were transfected with 1 μg β-Catenin-LEF1 or empty plasmid plus 1 μg TOPFlash, FOPFlash (Sigma Aldrich), or a RBPJ reporter plasmid and 20 ng pRL-TK plasmid (Promega) by using Lipofectamine 2000 according to the manufacturer's instructions (Invitrogen). Twenty four hours after transfection, 100,000 cells/ml were seeded as spheres in 24-well plates or reseeded in 6-well plates under adherent conditions. Two days after transfection cells were lysed and firefly and renilla luciferase activities were assayed according to the manufacturer's instructions (Promega). Three technical replicates per primary cell culture and group were analyzed. Knockdown efficiency was analyzed by western blotting and RT-qPCR.

Organoid cultures were transfected with 1 μg β-Catenin-LEF1, MAML1, NICD or empty plasmid plus 1 μg TOPFlash, FOPFlash (Sigma Aldrich), or a RBPJ reporter plasmid and 20 ng pRL-TK plasmid (Promega) by using Lipofectamine 2000. Briefly, organoids were dissociated with TrypLE. Single-cell suspensions were mixed with transfection complexes in 48-well plates, spun down at 600g and incubated at 37 °C for 4 h. Cells were seeded in Matrigel as described above and 48 h after transfection cells were lysed and firefly and renilla luciferase activities were assayed according to the manufacturer's instructions (Promega).

Adherent cultures were transfected with 2 μg TOPFlash, FOPFlash (Sigma Aldrich), or a RBPJ reporter plasmid and 20 ng pRL-TK plasmid (Promega) by

using Lipofectamine 2000 according to the manufacturer's instructions (Invitrogen). Twenty four hours after transfection, 100.000 cells/ml were seeded as spheres in 24-well plates or reseeded in 6-well plates under adherent conditions. Two days after transfection, cells were lysed and firefly and renilla luciferase activities were assayed according to the manufacturer's instructions (Promega). Three technical replicates per primary cell culture and group were analyzed.

**ELISA.** Supernatants of sphere, organoid and adherent cultures were collected after 7 days of culture. Elisa for WNT7B and WNT10A (Dianova) was performed according to the manufacturer's instructions using 100 μl of undiluted supernatant. Protein concentrations were calculated from WNT10A and WNT7B standard curves and were normalized to the number of cells per ml medium. Values are represented as ng protein per $10^6$ cells. Two technical replicates were measured per sample.

**Inhibitor assays.** Cells were seeded at 100.000 cells/ml in 24-well plates under non-adherent conditions and treated with ICG-001, DAPT, XAV-939, C59, IMR-1, and LF3 at the concentrations indicated in the Results for seven days. Medium was changed after 3 days. Spheres > 25 μm were counted per well and sphere number was normalized to number of spheres in untreated control.

Cells were seeded at 5.000 cells/ml in 9 μl Matrigel in 96-well plates and treated with ICG-001, DAPT, XAV-939, C59, IMR-1, and LF3 at the concentrations indicated in the results for seven days. Medium was changed every other day. Metabolic activity was measured by Celltiter Glo assay (Promega) according to the manufacturer's instructions. Three technical replicates per concentration were analyzed and values were normalized to non-treated controls.

**Xenograft assay.** All animal experiments were performed by EPO (Experimental Pharmacology and Oncology, Berlin-Buch). Animal experiments were carried out in accordance with the of the German Animal Protection Law and approved by the local responsible authorities. EPO complies to the EU guideline "European convention for the protection of vertebrate animals used for experimental and other scientific purposes. (EST 123)". Further, we handle our animals according to the "Regulation on the protection of experimental scientific purposes or other Purposes used animals". Compliance with the above rules and regulations is monitored by the Landesamt fuer Gesundheit und Soziales (LAGeSo) which is the responsible regulatory authority monitoring the animal husbandry based on the German Animal Welfare Act. Approval was given after careful inspection of the site including bedding, feeding & water, ventilation, temperature, and humidity, cleaning and hygiene concepts. Subcutaneous PDXs were established from 4 × 4 mm ccRCC tissue cubes in nude (Nu/J) mice and all subsequent experiments were performed in NSG (NOD.Cg-Prkdcscid Il2rgtm1Wjl/SzJ, NOD scid gamma) mice. Specimens of ccRCC patients with distant metastases were used to enhance engraftment efficiency[14]. Tumor growth was monitored for 3 months. Successfully engrafted tumors were re-passaged and expanded for inhibitor treatment. Mice were treated with 10 mg/kg DAPT i.p every 3 days or 100 mg/kg ICG-001, i.p. three times a week. Tumor volumes were measured with a caliper. Mice were sacrificed when control tumors reached 1 cm³ in volume.

Subcutaneous xenografts were harvested when tumors reached 1 cm³. Single cells isolation and FACS was performed as described above. $CXCR4^+MET^+CD44^+$ and $CXCR4^-MET^-CD44^-$ cells were diluted to appropriate concentrations and injected orthotopically into the kidney parenchyma. Mice were observed for up to 3 month's or until palpable tumors were formed in at least one group and sacrificed. Kidneys were collected and analyzed for tumor formation.

**Immunohistochemistry and immunofluorescence.** Tissue samples were fixed in 10% neutral buffered formalin overnight at 4 °C, dehydrated, embedded in paraffin and cut into 5 μM sections. Prior to staining, slides were rehydrated. For immunohistochemistry, antigens were retrieved by boiling sections in Tris-EDTA buffer (pH 9.0). Sections were incubated overnight at 4 °C with Ki-67 (1:200, Thermo Fisher), NOTCH1 (1:200, Rockland Inc.), β-Catenin (1:500, Cell Signaling Technology), Calbindin (1:3000, Sigma Aldrich) primary antibodies and horseradish peroxidase-conjugated secondary antibody. Staining was developed using the Envision + kit according to the manufacturer's recommendations (Dako). Sections were counterstained with eosin and photos were taken with an Axioscop (Zeiss) and the Axiocam Hrc (Zeiss) using the Zen software (Zeiss). Ki-67 staining in ccRCCs and xenografts was scored from 0 to 3, with 0 meaning no positive cells and 3 meaning that the tumor cells were Ki-67 positive.

For immunofluorescence, antigens were retrieved by boiling sections in Tris-EDTA buffer (pH 9.0). Section were incubated overnight at 4 °C with antibodies against CA9 (1:200, Abcam), CD10 (1:30, Dako), CD44 (1:400, BD biosciences), CXCR4 (1:200, Abcam), MET (1:50, Cell Signaling Technology), VCAM1 (1:250, Abcam), E-Cadherin (1:1000, BD biosciences), Epcam (1:500, Abcam), Ki-67 (1:200, Thermo Fisher), or JAG1 (1:200, Atlas Antibodies) primary antibodies and Alexa-conjugated secondary antibodies (1:200, Dianova). In case stainings were performed with more than one antibody of the same host species, we used a sequential protocol. Briefly, sections were incubated with primary antibodies o/n followed by Alexa-conjugated secondary antibodies (1:250) for 1.5 h, then blocked and incubated with the second set of primary antibodies o/n followed by the second

set of Alexa-conjugated secondary antibodies (1:200) for 30 min. To mark proximal tubules, sections were incubated with biotinylated LTL (1:200, Vector laboratories) at 4 °C overnight. Nuclei were counterstained with DAPI (100 ng/ml) and photos were taken with an Axios Imager (Zeiss) and the Axiocam MRm (Zeiss) using the Zen software (Zen). An overview of all antibodies in this study is provided in Supplementary Table 6.

For PAS staining, slides were incubated 5 min in 0.5% periodic acid followed by Schiff reagent for 20 min at RT. Sections were counterstained with eosin. For hematoxylin-eosin (HE) staining, slides were incubated 2 min in hematoxylin. Sections were counterstained with eosin fro 5 min. Photos were taken with an Axioscop (Zeiss) and the Axiocam Hrc (Zeiss) using the Zen software (Zeiss).

**In situ hybridization.** For in situ hybridization, Axin2 probe was linearized by SallI and labeled using the DIG RNA Labeling Mix (Roche). Sections were rehydrated, refixed, bleached, digested with proteinase K and acetylated with acetic anhydride. Hybridization was performed o/n at 63 °C. Sections were blocked and incubated with anti-DIG Fab (1:1000, Roche) o/n and stained with BM purple AP substrate for 24 h.

**Transmission electron microscopy (TEM).** Organoids were fixed in 2% (w/v) formaldehyde and 2.5% (v/v) glutaraldehyde in 0.1 M phosphate. After embedding in 10% agarose, samples were post-fixed with 1% (v/v) osmium tetroxide, dehydrated in a graded series of EtOH, and embedded in PolyBed® 812 resin. Ultrathin sections (60–80 nm) were stained with uranyl acetate as well as lead citrate and examined at 80 kV with a Zeiss EM 910 electron microscope. Acquisition was done with a Quemesa CCD camera and the iTEM software (Emsis GmbH).

**Descriptive statistics, significance and power calculations.** All statistical analyses were performed in GraphPad Prism (Graph Pad) and R unless stated otherwise. D'Agostino and Pearson omnibus test, t-test or Mann–Whitney-U-test, one-, RM, and two-way ANOVA with Tukey and Dunnett's correction for multiple comparison, and Spearman correlation were used. IC50 was calculated by non-linear regression analysis of the normalized response and the log of the inhibitor concentration fitting a curve with a variable slope. All tests were performed two-sided and a p-value < 0.05 was considered statistically significant. Kaplan–Meyer and Cox regression analyses were performed in R and SPSS (IBM). For xenograft assays, we estimated that we would need at least three samples per treatment group to see a two-fold change in tumor volume, for a power of 80% and for the probability of type I error (α) = 0.05. Power calculations were performed in G*Power 3.

**Reporting summary.** Further information on research design is available in the Nature Research Reporting Summary linked to this article.

## Data availability
The Illumina bead chip experiments (data shown in Fig. 4, Supplementary Fig. 4, and Supplementary Data 1) has been deposited in the Gene Expression omnibus under accession number GSE89461, the gene expression data from fresh frozen ccRCC tissues (data shown in Fig. 4 and Supplementary Fig. 4) is deposited under accession code GSE66270 and GSE66271, the single-cell sequencing (data shown in Fig. 5 and Supplementary Fig. 5) has been deposited under accession code GSE110680. All other data is available in the Article, Supplementary Information or available from the authors upon reasonable request.

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

## Acknowledgements

The authors thank Dr. Hans-Peter Rahn (MDC, Berlin) for teaching expertise in FACS, Dr. Samantha Praktiknjo (BIMSB, Berlin) for advice on single-cell sequencing, Dr. Silke Radetzki and Dr. Jens von Kries from the BIH Core Facility "Chemical Biology" for advice with the inhibitor assays, Miriam Heuer for experimental help, Dr. Ergin Kilic for help with the histology, and Russ Hodge (MDC, Berlin) for helpful discussions and suggestions for improving the text. This work was supported by a research grant of the Berlin Institute of Health (BIH). A. F. was funded by the Urological Research Foundation Berlin, head Prof. Stefan Loening.

## Author contributions

A.F. and W.B. developed the concept of the paper and wrote the manuscript. A.F. designed and performed experiments and analyzed the data. A.F., D.B., and A.M. performed experiments on organoids. A.W.-G. performed xenograft experiments. K.S. performed single-cell sequencing. A.F. conducted bioinformatics analysis. S.J. and B.E. helped with experimental work. S.K., D.B., and A.F. performed TEM experiments. J.B. and S.E. provided clinical specimens and expertise. J.B., W.C., and K.J. reviewed and discussed results and contributed to the preparation of the manuscript. W.B. supervised the project.

## Competing interests

The authors declare no competing interests.

## Additional information



