## [Peer Review File · Nature Communications]

Reviewers' comments:

Reviewer #1 (Remarks to the Author): Expert in kidney cancer and signalling

In this study, the authors characterize a subpopulation of ccRCC cells isolated based on expression of three putative stem cell markers, CD44, MET and CXCR4. They show that the frequency of these cells correlates to adverse prognosis and that they displayed enhanced sphere-forming capacity and growth in a subcutaneous mouse model system. They also developed a system for ccRCC organoids. Genome wide profiling of triple positive cells displayed enhanced WNT and NOTCH signaling activity and pharmacological inhibition of these pathways attenuated sphere forming capacity and growth of PDX tumors.

General comments:

1. It is unclear how the three markers (CD44, MET and CXCR4) were selected. The authors claim that CD44 is a novel stem cell marker in ccRCC, however it has been implicated in several publications (for review see Corro and Moch see *J Pathol Clin Res*, 2018;4:3–18). Also, ccRCC organoid cultures have been reported before and should be referenced (e.g. Grassi et al. *Cell Death and Disease* (2019)10:201).
2. The FACS plots show that the CD44 signal display a graded expression and it might be wrong to call the cells CD44 positive (or negative). If anything, it should be termed a CD44 high cell population.
2. In the paper there is a lack of staining for the selected markers in the primary tumors. It would be of great importance to clarify if the triple marker can identify a positive subpopulation of tumor cells in primary tumors and/or PDX tumors. In Figure 1F there is an effort to quantitate the expression level of the three markers in xenograft tumors, but the resolution of the picture is too poor to allow for identification of triple positive cells.

Specific comments:

1. Figure 1C: The authors claim that spheres formed by the triple-positive cells were larger. This size-difference should be quantified.
2. Figure 1E: The number of mice transplanted with tumor cells is too low for an assessment of grafting efficiency.
3. In Figure 2 the authors analyze the number of triple-positive cells in sphere and organoid cell cultures. They show that there is a clear enrichment of triple-positive cells when the cells are grown under sphere forming conditions, while organoid cultures lead to a depletion of positive cells. Staining with CA9 indicates that the structures are composed of tumor cells, and that E-cadherin is upregulated in organoid but not sphere cultures. These structures should be stained with CD44, MET and CXCR4 to clarify the distribution of positive cells in the respective growth conditions.
4. In Figure 3A the authors present data from a microarray analysis of control, sphere and FACS sorted cells. It is unclear how these experiments were performed, and how the data obtained was processed.
5. In Figure 3O the authors provide data based on analyses of the TCGA ccRCC cohort regarding overall survival in relation to high or low DKK3 and NOTCH3 expression. Analyses of WNT and NOTCH pathway signatures would be more informative than particular genes in the respective pathways.
6. In Figure 5J and Figure S3 data is presented indicating that WNT signaling induces Notch signaling, while NOTCH signaling is unable to regulate WNT signaling. However, this is based on ectopic activation of WNT signaling using beta-catenin-LEF1 fusion protein, while the corresponding Notch experiment is based on NOTCH1-siRNA transfection. There is a pronounced redundancy between NOTCH receptors and hence targeting only one receptor will not be sufficient. Overexpression of a constitutively active NOTCH receptor or overexpression of a dominant negative MAML construct would be preferable in this experiment.
7. There is a lack of information on how the authors calculated the stem cell signature (Figure 3O and Figure S3H).
8. Regarding the single cell data (Figure 4) it is very difficult to understand how the experiment

were performed and how the data was processed. In addition, the authors identify three clusters, but the characterization of these clusters is incomplete and provide limited information regarding the nature of the CSCs.

9. In Figure 5 and 6 the authors use pharmacological inhibitors of WNT and NOTCH signaling in vitro and they show that the responses could be classified into four groups, ICG-001, DAPT, ICG-001/DAPT and non-responders. It would be of importance to clarify whether these response patterns relates to the basic characteristics of the tumors, including clinical parameters and frequency of CD44/MET/CXCR4 cells.

Reviewer #2 (Remarks to the Author): expert in single cell sequencing

The paper is, what it is. Another claim to have found a cancer stem cell.

Specific criticism:

1) I am not clear why you chose MET as a marker. This is a fundamental bias of your study. Yes, some ccRCC do express MET, which however is more strongly associated with a different type of renal cancer, papillary RCC.

2) How do your cells compare to VCAM1+ proximal tubular cell that have recently been identified to be the "cellular identity" of ccRCC (PMID 30093597)?

3) What cell type from the above paper do your cells resemble?

4) I do not think the paper is terribly well written. I would suggest you asked a native speaker to edit it. An example of imperfect language would be the sentence: "But obviously, the...".

Reviewer #3 (Remarks to the Author): Expert in organoids

This manuscript documents the identification of a novel candidate population of human kidney cancer stem cells characterized by co-expression of CXCR4, MET and CD44. Elevated numbers of these CSC's in human kidney cancers are associated with poor prognosis. Sorted triple positive cells exhibit potent stem cell identity in sphere-forming assays and 3D organoid assays and accordingly exhibit elevated tumor forming capacity in xenotransplant assays. Expression profiling analyses document phenotypic heterogeneity within this population, with elevated levels of WNT and NOTCH signaling. Pharmacological inhibition of Wnt signaling in ex vivo culture and PDX models impairs tumor formation with a concomitant reduction in triple positive cells, hinting at its clinical relevance.

This study makes some interesting observations with significant clinical potential. However, the study suffers from a lack of consistency, with different PDX samples/target genes being used/analysed in different experiments, making it difficult to judge overall reproducibility of the

major findings. The drug treatment assays are also difficult to judge because of a failure to include non-cancer kidney cells/organoids to control for general toxicity.

Major critique

- 1) Fig 1c – please properly quantify this and include more than a single example to emphasize consistency of the result – does this merely reflect a difference in the proliferation status of the 2 populations (or different growth factors requirements ex vivo)?
- 2) Fig 1e – The numbers of individual transplantation experiments performed per sample are low. Can these be expanded to n=3 for each dilution to ensure reproducibility? Why was PDX4, used in the drug treatment experiments, not included here? What is the difference in tumor forming capacity of the primary (uncultured) triple positive versus triple negative populations (subcutaneous)?
- 3) Fig 1f – Do the PDX models also recapitulate the primary tumors in terms of % of triple positive cells, proliferation status, metastatic potential etc? Inclusion of a more comprehensive panel of lineage markers (podocyte markers, distal nephron markers for example), proliferation markers etc throughout the manuscript would have been helpful to properly evaluate PDX/tumor/organoid phenotypes.
- 4) Fig 3 – what were the selection criteria for the ccRCC samples included in the microarray analyses? Were cells from the same patients used in any of the PDX models etc? It was unclear to me what the “control” cells were – are these the adherent cells (presumptive non stem cells) from the sphere cultures or triple negative primary cells? Please clarify and justify the choice of control sample. To my knowledge, Lgr4 has not been shown to be responsive to Wnt signaling, but is instead involved in regulating Wnt signaling activation at the cell membrane. Given that Lgr4 is one of the few genes in the manuscript referred to as being a Wnt target that responds to the pharmacological inhibition assays, it is important to clarify this point.
- 5) Fig 3a – why are the ccRCC10 cultured versus primary cells demonstrating such different expression profiles? In fig 3 and suppl fig 3, it would greatly increase confidence in the reproducibility of the findings if the same expanded panel of Wnt/Notch targets was used in all figure panels. If the panel of gene expression changes are not reproducible for different samples, then this should be acknowledged and discussed.
- 6) Fig 3i,j – I believe that the reporter gene assays were performed on sphere cultures from 5 different patient samples? – if so, which samples were chosen and why? Did they differ in their aggressiveness, metastatic potential etc?
- 7) Fig 4 – It appears that only 2 samples were included in this single cell RNAseq analysis – what were the selection criteria? Would the heterogeneity be different amongst samples displaying divergent metastatic potential (ie, different degrees of aggressiveness. Do the subsets display different proliferation status? Is NOTCH3 and WNT10A expression different amongst the subsets? Do the different subsets display similar sensitivity to the Wnt/Notch inhibitors?
- 8) Some validation of the above findings in primary human kidney cancer sections would be useful – for example, is Wnt10b or Wnt/Notch target genes differentially expressed in cancer versus normal tissue?
- 9) Fig 5 - Again, the panel of Wnt/Notch target genes evaluated here should be expanded and matched with those used in other assays to ensure reproducibility. Given that Wnt10A is upregulated in cultures enriched for the CSC's, it seems reasonable to assume that the pathway is being activated in kidney cancer at the membrane. Does treatment with Porcupine inhibitors then block the observed Wnt/Notch signaling activity and phenocopy the growth inhibition on cultured spheres.
- 10) Suppl 5e – the conclusion that DAPT treatment is selectively impacting the CSC's in the organoids is overstated – is there a reduction in the number of triple positive cells after treatment? Apoptosis evident? Is this selective, with no similar effects on non cancer kidney organoids?
- 10) Fig 6 – the effects of the Wnt/Notch inhibitors on the spheres/PDX tumors is potentially very interesting. However, I remain somewhat unconvinced of the selectivity of the drugs being used. Please include a non cancer control. There also appears to be no nuclear b-catenin evident in the tumor samples as one might expect given the Wnt pathway activation status. As previously

mentioned, Lgr4 might not be the most appropriate indicator of Wnt pathway status – please include a larger panel here. Is there apoptosis evident in the treated samples? Would Porcupine inhibitors work here? Why were PDX1 and PDX4 selected for this experiment in place of PDX1-3 as used in figure 1?

Response to the referees

We have carefully read all comments from the reviewers answered their questions, and have added new experiments to the revised version of the manuscript.

For clarity, all reviewer comments below are shown in blue, with our responses in black below. Novel parts of the manuscript that were included in the response are highlighted in yellow here as well as in the revised version of the manuscript.

We have worked extremely hard to improve our manuscript, and we hope that our responses will satisfy you and the reviewers.

Reviewer #1 (Remarks to the Author): Expert in kidney cancer and signalling

In this study, the authors characterize a subpopulation of ccRCC cells isolated based on expression of three putative stem cell markers, CD44, MET and CXCR4. They show that the frequency of these cells correlates to adverse prognosis and that they displayed enhanced sphere-forming capacity and growth in a subcutaneous mouse model system. They also developed a system for ccRCC organoids. Genome wide profiling of triple positive cells displayed enhanced WNT and NOTCH signaling activity and pharmacological inhibition of these pathways attenuated sphere forming capacity and growth of PDX tumors.

This is a correct description of our work.

General comments:

1. It is unclear how the three markers (CD44, MET and CXCR4) were selected. The authors claim that CD44 is a novel stem cell marker in ccRCC, however it has been implicated in several publications (for review see Corro and Moch see J Path: Clin Res, 2018;4:3–18). Also, ccRCC organoid cultures have been reported before and should be referenced (e.g. Grassi et al. Cell Death and Disease (2019)10:201).

The markers were selected on the basis of their description as markers for stem cells in the kidney, renal cancer stem cells, or cancer stem cells in other solid tumors. We have added the following sentence to the manuscript to clarify the selection process for the tested markers (see page 5 of the revised manuscript):

The surface markers were selected for having been previously identified as stem cell markers in the kidney (i.e. CD24, CD29, CD133)²⁴, and their distinctive appearance on cancer stem cells in other tissues (CD24, CD29, Epcam, CD44, MET, CD90, ALDH1A1 activity)²⁵⁻³⁰, or in the kidney (CD133, CD24, CD105, CXCR4)^{16,17,24,31}.

We have also specified our comment that CD44 has been used for the first time to sort cancer stem cells of freshly isolated human primary RCC cells. The reviewer is correct that its expression has been described on spheres or cells sorted with other markers in RCC, although it has not been used for sorting RCC CSC previously. We have now added to the manuscript (page 5):

We found that CD44 can further refine this population: CD44 is an integrin receptor that has been identified as a stem cell marker in a number of other cancer entities and was

shown to be elevated in other CSC populations in ccRCC^{16,17,30,36} but had not been previously used to sort CSC in ccRCC.

We have also added a discussion point about RCC organoids (page 17, the reference Grassi et al. is now added, ref. 68):

Two other methods to culture ccRCC organoids have recently been described^{72,73}. The first yielded organoids with a limited expansion capacity and a low rate of success. Those organoids showed less epithelial polarization than the organoids we produced here. Interestingly both previous methods described a maintenance of genomic clones in the organoids, supporting the assumption that organoids are representative of the primary tumors. Future work will be needed to clarify the differences between the cultivation methods and their significance.

2. The FACS plots show that the CD44 signal display a graded expression and it might be wrong to call the cells CD44 positive (or negative). If anything, it should be termed a CD44 high cell population.

The reviewer is correct, that CD44 populations are not clearly separated and that we observe a graded expression. We have however sorted all cells with positive CD44 expression in comparison to a negative control. The term CD44^{high} would suggest that a second CD44-positive population with lower expression exist, which is not the case. This is why we prefer to keep the term CD44⁺.

2. In the paper there is a lack of staining for the selected markers in the primary tumors. It would be of great importance to clarify if the triple marker can identify a positive subpopulation of tumor cells in primary tumors and/or PDX tumors. In Figure 1F there is an effort to quantitate the expression level of the three markers in xenograft tumors, but the resolution of the picture is too poor to allow for identification of triple positive cells.

Thanks for this helpful remark. We have now performed stainings for all 3 markers in 40 ccRCC cases. All of them were either used for FACS experiments and/or xenografts. We have stained simultaneously for CXCR4, MET, CD44 and VCAM. In addition, we stained for LTL, Calbindin and Ki67 on consecutive slides (Fig. 2, 3 and Supplementary Fig. 2, 3). We have detected CXCR4⁺MET⁺CD44⁺ cells of all specimens, but could not observe any preferential localisation (e.g. at the invasive front). Rather, we have found single cells located in all areas within the tumors. The frequency of the cells was low, as expected from the FACS experiments. The low frequency makes accurate scoring challenging, therefore we only used FACS data for quantifications. We have now added the following sentences to the manuscript (page 6):

Immunofluorescence established that PDXs were positive for both CA9 and CD10 (Supplementary Fig. 2b), confirming their identity as ccRCCs. In addition, Ki-67 scores were similar to or higher than the scores for the corresponding primary tumors in subcutaneous or orthotopic PDX (Supplementary Fig. 2c).

Immunofluorescence for CXCR4, MET and CD44 confirmed that CXCR4⁺MET⁺CD44⁺ cells are rare in ccRCC tumors and subcutaneous PDX (Fig. 2d). They remained low in the orthotopic xenografts, indicating that transplanted CXCR4⁺MET⁺CD44⁺ cells differentiated during tumor formation and lost the expression of these surface markers. We observed no preferential location of CXCR4⁺MET⁺CD44⁺ cells within the tumors, even though there were marked intra- and inter-patient differences in the expression of single markers. Few CD44-positive cells were detected in areas with a predominance of nested clear cells and more pronounced in areas that were more solid and

dedifferentiated (Supplementary Fig. 2d). MET was often strongly positive at tumor edges and more diffusely in the centers (Supplementary Fig. 2e). CXCR4 expression was often detected in single cells, rather than clusters of cells, throughout the tumor (Fig. 2d).

VCAM1, which has been proposed to mark the cell-of-origin in ccRCC⁴⁰, generally overlapped with MET in primary tumors and xenografts (Fig. 2e), but it was also expressed in cells besides CXCR4⁺MET⁺CD44⁺. Nevertheless, CXCR4⁺MET⁺CD44⁺ were in the vast majority positive for VCAM1, suggesting that the latter might represent a subpopulation of VCAM1⁺ cells in ccRCC. We further stained with lotus tetragonolobus lectin (LTL) and Calbindin, to explore the maintenance of proximal and distal tubule characteristics in the tumors. We detected LTL-positive cells in all tumors, even though the number of LTL-positive cells varied, but were unable to detect any Calbindin-positive tumor cells (Supplementary Figure 2f, g). Specificity of both markers was confirmed in normal adjacent tissue (Supplementary Figure 2h, i). This confirms RNA sequencing data suggesting that ccRCC maintains the expression features of proximal tubule cells^{2,40-43}.

We have extended these stainings to all xenografts, as well as sphere and organoid cultures (Fig. 3 and Supplementary Fig. 3). The xenograft stainings confirm that the CSC content is maintained in subcutaneous and orthotopic xenografts. The stainings in spheres and organoids confirm the FACS experiments, where spheres displayed a higher CSC content than organoids. We added the following sentences to the revised manuscript (page 8);

In sphere cultures, cells aggregated into solid structures (Fig. 3d). This contrasted with organoid cultures, where the majority of cells formed large hollow cysts (Fig. 3e). A subset of organoids exhibited other morphologies, such as more solid structures or intertwined tubes (Supplementary Fig. 3e). Such differences were observed in organoids derived from single patients, as well as from different individuals. Immunofluorescence staining in spheres revealed weak E-cadherin (ECAD) with no preferential association to the cell surface (Fig. 3f, upper left). In contrast, in organoids E-cadherin located to lateral cell membranes (Fig. 3g, upper left) indicating epithelial cell differentiation. Carbonic anhydrase IX (CA9) staining confirmed that both types of cultures consisted of kidney cancer cells (Fig. 3f, g, upper right)⁴⁶. LTL, which can be used to mark proximal tubule brush borders, only marked organoid cells (Fig. 3f, g, middle left). Although LTL staining was generally diffuse, it was localized apically in some of the organoids, which is typical for proximal tubules (Supplementary Fig. 3f). Most sphere cells were positive for CXCR4, MET, and CD44, but only a subset of cells of the organoids was positive for all three markers further indicating differentiation of organoid cells (Fig. 3f, g, middle right and Supplementary Fig. 3h). VCAM1-positive cells were detected in spheres and organoids (Fig. 3f, g, lower left), and both sphere and organoid cultures contained Ki-67-positive cells (Fig. 3f, g, lower right).

Specific comments:

1. Figure 1C: The authors claim that spheres formed by the triple-positive cells were larger. This size-difference should be quantified.

We have added the quantification in Figure 1d.

2. Figure 1E: The number of mice transplanted with tumor cells is too low for an assessment of grafting efficiency.

We agree that the results in Figure 2b (Figure 1e in the old version of the manuscript) are limited due to the small number of samples. In our hands, xenografts of ccRCC grew slowly, with 3 month latency in comparison to the formation of subcutaneous tumors. We also did not want to passage tumors multiple times to ensure that they still maintain the characteristics of the primary tumor. Therefore the material for our experiments was limited, especially as our population of interest is small (4.5 to 11.8% in case of the xenografts). Given that we still wanted to test more than one cell number, it was not possible to perform these experiments with 3 technical replicates. Unfortunately, repeating these experiments would take considerably longer than the time for review. We estimate the experiments to take 9 month to a year, as all xenografts need to be re-established and expanded from frozen tissues. Therefore we will not be able to repeat the experiments at this point and appreciate the reviewer's understanding.

3. In Figure 2 the authors analyze the number of triple-positive cells in sphere and organoid cell cultures. They show that there is a clear enrichment of triple-positive cells when the cells are grown under sphere forming conditions, while organoid cultures lead to a depletion of positive cells. Staining with CA9 indicates that the structures are composed of tumor cells, and that E-cadherin is upregulated in organoid but not sphere cultures. These structures should be stained with CD44, MET and CXCR4 to clarify the distribution of positive cells in the respective growth conditions.

Thanks for this remark. We have added the stainings in Figure 3. We generally observe more triple-positive cells in the spheres in contrast to organoids, confirming the results from FACS staining. We write in the revised manuscript (page 8):

Most sphere cells were positive for CXCR4, MET, and CD44, but only a subset of cells of the organoids was positive for all three markers further indicating differentiation of organoid cells (Fig. 3f, g, middle right and Supplementary Fig. 3h).

4. In Figure 3A the authors present data from a microarray analysis of control, sphere and FACS sorted cells. It is unclear how these experiments were performed, and how the data obtained was processed.

We have added a section in the Material and Methods to further clarify how the microarray experiments have been performed. We write on page 22:

RNA was isolated from uncultured and unsorted control cells, CXCR4⁺MET⁺CD44⁺ or sphere-cultured cells using the RNeasy Mini Kit (Qiagen) and hybridized to the HumanHT-12 v4 bead chip (Illumina Inc.) according to the manufacturers protocol. HumanHT-12 v4 bead chip data were normalized and log-transformed in Partek Genomics Suite (Partek) and analyzed using the limma package⁸⁷ in R. We identified a stem cell signature that included all genes that were upregulated in either spheres and CXCR4⁺MET⁺CD44⁺ with FC >1.5 or < -1.5 and p-value <0.05.

5. In Figure 3O the authors provide data based on analyses of the TCGA ccRCC cohort regarding overall survival in relation to high or low DKK3 and NOTCH3 expression. Analyses of WNT and NOTCH pathway signatures would be more informative than particular genes in the respective pathways.

We have analysed the WNT and NOTCH gene signature (this included all WNT and NOTCH genes displayed in Figure 4b). We do observe a survival benefit for patients

with low expression of these genes, which is in line with the results obtained using the cancer stem cell signature or single genes (see Fig. 4m). We now write on page 11:

The stem cell signature (Fig. 4l), the combination of all deregulated WNT and NOTCH pathway genes (Fig. 4m), or DKK3 (Fig. 4n) and NOTCH3 (Fig. 4o) expression alone were associated with the overall survival of the 462 patients.

6. In Figure 5J and Figure S3 data is presented indicating that WNT signaling induces Notch signaling, while NOTCH signaling is unable to regulate WNT signaling. However, this is based on ectopic activation of WNT signaling using beta-catenin-LEF1 fusion protein, while the corresponding Notch experiment is based on NOTCH1-siRNA transfection. There is a pronounced redundancy between NOTCH receptors and hence targeting only one receptor will not be sufficient. Overexpression of a constitutively active NOTCH receptor or overexpression of a dominant negative MAML construct would be preferable in this experiment.

We have overexpressed MAML1 and NICD and performed luciferase reporter gene assays to study WNT or NOTCH activation. The results from these experiments confirm that NOTCH activation does not impact WNT signalling activation. We write on page 10:

In sphere cultures, we further observed that when WNT signaling was ectopically activated using a β -catenin-LEF1 fusion protein⁵¹ (Fig. 4i), this triggered a strong induction of NOTCH signaling (Fig. 4j). In contrast, MAML, NICD, or NOTCH1 siRNA transfections had no significant effects on WNT signaling (Supplementary Fig. 4f-h).

The results were included in Supplementary Figure 4f-h.

7. There is a lack of information on how the authors calculated the stem cell signature (Figure 3O and Figure S3H).

We have included all genes from the microarray profiling of FAC-sorted cells and spheres with a FC >1.5 or <-1.5 and a p-value 0.05 in in either spheres or FAC-sorted cells. This information has been added to the Results and to Material and Methods.

8. Regarding the single cell data (Figure 5) it is very difficult to understand how the experiment were performed and how the data was processed. In addition, the authors identify three clusters, but the characterization of these clusters is incomplete and provide limited information regarding the nature of the CSCs.

The top-scoring genes in each cluster mainly contained genes encoding for extracellular, matrix, and transmembrane proteins and did not include any known stemness genes. We further saw that while some WNT or NOTCH genes and some stem cell genes were not among the top-scoring ones, they were still higher in Cluster 1 and 3, with other stem cell genes not differentiating between clusters. We conclude that all three clusters contain genes with stem cell characteristic, but these characteristics were highest in cluster 1 and 3. To explore the functional differences of each cluster as indicated by the differential expression of extracellular and transmembrane proteins would, as we believe, exceed the scope of this paper, albeit being an interesting finding. We now write (page 11):

This revealed three clusters (Fig. 5a, marked by different colors), which were not unique to individual patients, as cells from each sample were found in each of the clusters. We identified the top 20 genes of each cluster (Fig. 5b, indicated on the right). The genes included extracellular and transmembrane genes and extracellular vesicles, such as

CXCL1, CXCL6, MMP7, CD70 and COL5A1. While the cancer stem cell genes and WNT and NOTCH signaling genes were not included in this list, we found that clusters 1 and 3 showed higher expression of WNT and NOTCH pathway genes, stem cell genes and kidney-specific genes, for instance LGR4, TCF7L2, JAG1, ALDH1A1 and PAX2 (Fig. 5c). Other stem cell-specific genes were expressed across all three clusters (Supplementary Fig. 5c). The results indicate that while cluster 1 and 3 have the highest expression of certain CSC genes, in general all clusters share expression of a subset of CSC genes. Interestingly, WNT and NOTCH genes are both activated in the same cell clusters and do not mark different subpopulations of cells.

9. In Figure 5 and 6 the authors use pharmacological inhibitors of WNT and NOTCH signaling in vitro and they show that the responses could be classified into four groups, ICG-001, DAPT, ICG-001/DAPT and non-responders. It would be of importance to clarify whether these response patterns relate to the basic characteristics of the tumors, including clinical parameters and frequency of CD44/MET/CXCR4 cells.

We have correlated the IC50 values for each specimen with the pathological stage and Fuhrman grade as well as the frequency of CXCR4⁺MET⁺CD44⁺ cells. We could not observe any significant correlations for either ICG-001 or DAPT. These results indicate that specific response mechanisms exist on a molecular basis that, as we believe, need have to be investigated in the future. We write on page 13:

Responses to either ICG-001 or DAPT did not correlate with the pathological stage or grade of the cancer or percentage of CXCR4⁺MET⁺CD44⁺ cells (Supplementary Table 5) indicating that specific mechanisms of response exist.

Reviewer #2 (Remarks to the Author): expert in single cell sequencing

The paper is, what it is. Another claim to have found a cancer stem cell.
1) I am not clear why you chose MET as a marker. This is a fundamental bias of your study. Yes, some ccRCC do express MET, which however is more strongly associated with a different type of renal cancer, papillary RCC.

MET was selected on the basis of previous observations that it is a suitable marker for cancer stem cells in other solid tumors (specifically in combination with CD44). In addition, MET expression has been associated with tumor aggressiveness in ccRCC, even though, as the reviewer pointed out, MET mutations are rare in ccRCC. We now write on page 5:

The surface markers were selected for having been previously identified as stem cell markers in the kidney (i.e. CD24, CD29, CD133)²⁴, and their distinctive appearance on cancer stem cells in other tissues (CD24, CD29, Epcam, CD44, MET, CD90, ALDH1A1 activity)²⁵, or in the kidney (CD133, CD24, CD105, CXCR4)^{16,17,24,26}.

2) How do your cells compare to VCAM1⁺ proximal tubular cell that have recently been identified to be the "cellular identity" of ccRCC (PMID 30093597)?

The study the reviewer is referring to suggests VCAM⁺ proximal tubules as the cell of origin in ccRCC, and the authors show that VCAM expression overlaps with CA9 expression in ccRCC tissue and predisposed normal cells with germline VHL mutation. We have here simultaneously stained for VCAM, CXCR4, MET, and CD44 in primary tumours and xenografts. We observed that a subset of cells in ccRCC cells are VCAM-positive and that the frequency of VCAM⁺ cells was considerably higher than the

frequency of CXCR4⁺MET⁺CD44⁺ cells. So, CXCR4⁺MET⁺CD44⁺ cells appear to be a subset of VCAM⁺ cells. We write on page 7:

VCAM1, which has been proposed to mark the cell-of-origin in ccRCC⁴⁰, generally overlapped with MET in primary tumors and xenografts (Fig. 2e), but it was also expressed in cells besides CXCR4⁺MET⁺CD44⁺. Nevertheless, CXCR4⁺MET⁺CD44⁺ were in the vast majority positive for VCAM1, suggesting that the latter might represent a subpopulation of VCAM1⁺ cells in ccRCC.

3) What cell type from the above paper do your cells resemble?

As mentioned above, Young et al. suggested that VCAM⁺ proximal tubule cells, more specifically from the S1 segment of proximal tubules, might be the cell of origin of ccRCC. Previous studies have used bulk RNA sequencing and likewise pinpointed to proximal tubules as the cells of origin. We stained for LTL (proximal tubules) and Calbindin (distal tubules) and found only LTL-positive cells in ccRCC tissue, confirming that ccRCC retains some characteristics of proximal tubules. In addition, our organoids are positive for LTL. As mentioned above, the CXCR4⁺MET⁺CD44⁺ cells were VCAM-positive. Therefore, we conclude that our Results fit with the reports from bulk and single cell RNA sequencing, but want to point out that the CXCR4⁺MET⁺CD44⁺ cells are a smaller subset of the cells described by Young et al. We write in the manuscript (page 7):

We further stained with lotus tetragonolobus lectin (LTL) and Calbindin, to explore the maintenance of proximal and distal tubule characteristics in the tumors. We detected LTL-positive cells in all tumors, even though the number of LTL-positive cells varied, but were unable to detect any Calbindin-positive tumor cells (Supplementary Figure 2f, g). Specificity of both markers was confirmed in normal adjacent tissue (Supplementary Figure 2h, i). This confirms RNA sequencing data suggesting that ccRCC maintains the expression features of proximal tubule cells^{2,40-43}.

4) I do not think the paper is terribly well written. I would suggest you asked a native speaker to edit it. An example of imperfect language would be the sentence: "But obviously, the...".

We sent the revised manuscript to the scientific writer of the MDC, Russel Hodge, an American who has a University degree in English language, for improvements of language and style. We hope that it now meets the reviewer's expectations.

Reviewer #3 (Remarks to the Author): Expert in organoids

This manuscript documents the identification of a novel candidate population of human kidney cancer stem cells characterized by co-expression of CXCR4, MET and CD44. Elevated numbers of these CSC's in human kidney cancers are associated with poor prognosis. Sorted triple positive cells exhibit potent stem cell identity in sphere-forming assays and 3D organoid assays and accordingly exhibit elevated tumor forming capacity in xenotransplant assays. Expression profiling analyses document phenotypic heterogeneity within this population, with elevated levels of WNT and NOTCH signaling. Pharmacological inhibition of Wnt signaling in ex vivo culture and PDX models impairs tumor formation with a concomitant reduction in triple positive cells, hinting at its clinical relevance.

This is an excellent description of the main Results of our paper.

This study makes some interesting observations with significant clinical potential. However, the study suffers from a lack of consistency, with different PDX samples/target genes being used/analysed in different experiments, making it difficult to judge overall reproducibility of the major findings. The drug treatment assays are also difficult to judge because of a failure to include non-cancer kidney cells/organoids to control for general toxicity.

1) Fig 1c – please properly quantify this and include more than a single example to emphasize consistency of the result – does this merely reflect a difference in the proliferation status of the 2 populations (or different growth factor requirements *ex vivo*)?

We do appreciate the comment. The results in Figure 1b are the combined results from 4 assays. The cells were grown under the same growth factor conditions, which are commonly used for sphere cultures. Sphere formation requires more than proliferation; for examples the potential for anchorage-independent growth and sphere formation are considered a functional assessment for stemness (see Dontu et al. Genes Dev, 2003, doi:10.1101/gad.1061803). If the differences were merely based on proliferation, we would expect the same number of spheres formed but they should be of smaller size. Yet, what we observed and present in Figure 1b is that 1:80 cells are able to form a sphere when seeding CXCR4⁺MET⁺CD44⁺, but 10-times more cells do so in the negative population. In addition, in organoids proliferation does not seem to be restricted to CXCR4⁺MET⁺CD44⁺ cells, and the number of proliferating cells in spheres and organoids did not differ (see Figure 3f and g).

2) Fig 1e – The numbers of individual transplantation experiments performed per sample are low. Can these be expanded to n=3 for each dilution to ensure reproducibility? Why was PDX4, used in the drug treatment experiments, not included here? What is the difference in tumor forming capacity of the primary (uncultured) triple positive versus triple negative populations (subcutaneous)?

As we have mentioned above, in response to reviewer 1, we do agree that the results from the xenografting experiments suffer from the limited amount of tissue that was available. We observed in this study that kidney tumor cells grow badly as PDX, in contrast to other tumor cells, on which our lab also has experience (see for instance Zhu et al., Cell Reports 2019). Primaries from metastasized kidney tumors therefore needed to be used. To fulfil the requirements of this reviewer, we would need 9 months or longer, which is considerably longer than our revision time. We appreciate the reviewers understanding.

3) Fig 1f – Do the PDX models also recapitulate the primary tumors in terms of % of triple positive cells, proliferation status, metastatic potential etc? Inclusion of a more comprehensive panel of lineage markers (podocyte markers, distal nephron markers for example), proliferation markers etc throughout the manuscript would have been helpful to properly evaluate PDX/tumor/organoid phenotypes.

Thanks for this comment. We have simultaneously stained for CXCR4, MET and CD44, and we find that the results are generally comparable between primary tumors,

subcutaneous xenografts and orthotopic xenografts (Fig. 2 and Supplementary Fig. 2). We also stained for Ki-67, and we do observe either the same or higher scores in comparison to the primary tumors. We have further included LTL and Calbindin staining of primary tumors, xenografts and cultures and found that the tumors are all positive for LTL although to varying degrees. They are also VCAM-positive, which is a specific marker of the S1 kidney segment. We write on page 7 in the revised manuscript:

Immunofluorescence for CXCR4, MET and CD44 confirmed that CXCR4⁺MET⁺CD44⁺ cells are rare in ccRCC tumors and subcutaneous PDX (Fig. 2d). They remained low in the orthotopic xenografts, indicating that transplanted CXCR4⁺MET⁺CD44⁺ cells differentiated during tumor formation and lost the expression of these surface markers. We observed no preferential location of CXCR4⁺MET⁺CD44⁺ cells within the tumors, even though there were marked intra- and inter-patient differences in the expression of single markers. Few CD44-positive cells were detected in areas with a predominance of nested clear cells and more pronounced in areas that were more solid and dedifferentiated (Supplementary Fig. 2d). MET was often strongly positive at tumor edges and more diffusely in the centers (Supplementary Fig. 2e). CXCR4 expression was often detected in single cells, rather than clusters of cells, throughout the tumor (Fig. 2d).

VCAM1, which has been proposed to mark the cell-of-origin in ccRCC⁴⁰, generally overlapped with MET in primary tumors and xenografts (Fig. 2e), but it was also expressed in cells besides CXCR4⁺MET⁺CD44⁺. Nevertheless, CXCR4⁺MET⁺CD44⁺ were in the vast majority positive for VCAM1, suggesting that the latter might represent a subpopulation of VCAM1⁺ cells in ccRCC. We further stained with lotus tetragonolobus lectin (LTL) and Calbindin, to explore the maintenance of proximal and distal tubule characteristics in the tumors. We detected LTL-positive cells in all tumors, even though the number of LTL-positive cells varied, but were unable to detect any Calbindin-positive tumor cells (Supplementary Figure 2f, g). Specificity of both markers was confirmed in normal adjacent tissue (Supplementary Figure 2h, i). This confirms RNA sequencing data suggesting that ccRCC maintains the expression features of proximal tubule cells^{2,40-43}.

4) Fig 3 – what were the selection criteria for the ccRCC samples included in the microarray analyses? Were cells from the same patients used in any of the PDX models etc? It was unclear to me what the “control” cells were – are these the adherent cells (presumptive non stem cells) from the sphere cultures or triple negative primary cells? Please clarify and justify the choice of control sample. To my knowledge, Lgr4 has not been shown to be responsive to Wnt signaling, but is instead involved in regulating Wnt signaling activation at the cell membrane. Given that Lgr4 is one of the few genes in the manuscript referred to as being a Wnt target that responds to the pharmacological inhibition assays, it is important to clarify this point.

Primary ccRCC cells were isolated during the whole course of this project, i.e. over three years. We have usually used fresh lines in our analysis, which had not been passaged more than 3 times. In case of the microarray experiments, the cells were not cultured before they were FAC-sorted and only cultured for a week as spheres to obtain the sphere samples. Therefore, samples were not preselected but chosen upon availability. Due to the low take rates for xenografts (see above) and the fact that we only established xenografts from primary tumors if distance metastases were already diagnosed, we could not find an overlap between the xenografts and the samples used for the microarray. The term control in the microarray experiments refers to non-cultured and unsorted tumor cells (see Material and Methods for the isolation protocol).

We have included this information in the revised manuscript for clarity. We write (page 9):

We carried out genome-wide expression profiling of FAC-sorted CXCR4⁺MET⁺CD44⁺, sphere and uncultured, non-sorted control cells from the tumors of three patients.

We have now also included CD44, LEF1 and Axin2 as WNT targets in all figures. For the additional WNT inhibitors, we only used LEF1 and AXIN2, as CD44 failed to be statistically significant. We have removed LGR4 and BIRC5, since the reviewer is correct to point out that LGR4 is not a WNT target

5) Fig 3a – why are the ccRCC10 cultured versus primary cells demonstrating such different expression profiles? In fig 3 and suppl fig 3, it would greatly increase confidence in the reproducibility of the findings if the same expanded panel of Wnt/Notch targets was used in all figure panels. If the panel of gene expression changes are not reproducible for different samples, then this should be acknowledged and discussed.

The reviewer is correct to point out that ccRCC varies from the ccRCC2 and ccRCC4, and we do not have a biological explanation for this. However, validation by RT-qPCR in 10 more tumors shows that upregulation of stemness, WNT and NOTCH genes was found in general. As mentioned above, we have now included CD44 and LEF-1 as WNT targets in the inhibitor experiments and excluded LGR4 and BIRC5.

6) Fig 3i,j – I believe that the reporter gene assays were performed on sphere cultures from 5 different patient samples? – if so, which samples were chosen and why? Did they differ in their aggressiveness, metastatic potential etc?

As mentioned above, we performed assays in cultures that we isolated over the entire course of the project. In all experiments the samples were chosen randomly and based on availability. Samples were not selected based on aggressiveness or metastatic potential, except for the generation of xenografts, where we preselected samples from patients with distant metastases, due to the reported enhanced engraftment efficiency (see also Pavia-Jimenez et al. Nat Protoc 2014, doi:10.1038/nprot.2014.108).

7) Fig 4 – It appears that only 2 samples were included in this single cell RNAseq analysis –what were the selection criteria? Would the heterogeneity be different amongst samples displaying divergent metastatic potential (ie, different degrees of aggressiveness? Do the subsets display different proliferation status? Is NOTCH3 and WNT10A expression different amongst the subsets? Do the different subsets display similar sensitivity to the Wnt/Notch inhibitors?

We FAC-sorted freshly isolated cells that had not been cultured before, which again means that the samples were chosen based on availability. To ensure that clustering would not be based on the differences between the samples, we combined both datasets and thus identified clusters that represented differences that were commonly found in both samples so that the cluster would not merely reflect sample differences.

8) Some validation of the above findings in primary human kidney cancer sections would be useful – for example, is Wnt10b or Wnt/Notch target genes differentially expressed in cancer versus normal tissue?

WNT10a has previously been reported as an oncogenic WNT ligand in ccRCC (Hsu et al. Plos One, 2012, doi:10.1371/journal.pone.0047649). Therefore we did not study its expression by Western blotting or IHC. The microarray experiments in Supplementary Figure 4 (Supplementary Figure 3 in the old version of the manuscript) have been performed on 28 cases with ccRCC. The Results show that the expression of the cancer stem cell signature differs between normal and tumor tissues. These included the WNT and NOTCH genes shown in Figure 4 (Figure 3 in the old version of the manuscript). We have further added a Kaplan-Meier plot in Figure 4m showing that the combined expression of all WNT and NOTCH genes (refer to Figure 4b for the full list of included genes) are associated with overall survival in the TCGA cohort.

9) Fig 5 - Again, the panel of Wnt/Notch target genes evaluated here should be expanded and matched with those used in other assays to ensure reproducibility. Given that Wnt10A is upregulated in cultures enriched for the CSC's, it seems reasonable to assume that the pathway is being activated in kidney cancer at the membrane. Does treatment with Porcupine inhibitors then block the observed Wnt/Notch signaling activity and phenocopy the growth inhibition on cultured spheres.

We have now included CD44 and LEF1 as WNT targets in all Figures. For the additional WNT inhibitors, we only used LEF1 and AXIN2, as CD44 failed to be statistically significant. We have tested the porcupine inhibitor C59 and it was inhibitory, but at lower efficiency as reported in other tumors. This may indicate that although we see enhanced secretion of WNT ligands, other downstream regulatory mechanisms control WNT signalling in ccRCC. We have not explored this further. In conclusion, ICG-001 appeared to be the most suitable WNT inhibitor for ccRCC in our experiments.

10) Suppl 5e – the conclusion that DAPT treatment is selectively impacting the CSC's in the organoids is overstated – is there a reduction in the number of triple positive cells after treatment? Apoptosis evident? Is this selective, with no similar effects on non cancer kidney organoids?

We have not observed any effect of DAPT on normal kidney cells in spheres or organoids (see Supplementary Figure 6). We have observed a reduction of CXCR4⁺MET⁺CD44⁺ cells after 1 to 7 days of DAPT treatment in spheres.

We did not stain for apoptotic markers. We write (page 14):

To exclude non-specific toxicity, we treated spheres and organoids derived from normal-adjacent tissue. DAPT had no significant effect on the 5 sphere and organoid cultures that were tested. ICG-001 produced no significant toxicity in sphere cultures, but led to a reduction of organoid growth.

To exclude that the effects of ICG-001 were due to unspecific events, we tested additional WNT inhibitors in specimens that responded to ICG-001. We used the TANKYRASE inhibitor XAV939⁵⁸ and the β -CATENIN inhibitor LF3⁵⁹, and the Porcupine inhibitor C59⁶⁰. While each of the WNT inhibitors reduced sphere formation and organoid growth, they showed different efficiencies, decreased the expression of WNT target genes to varying levels, and some had stronger effects on normal cells, indicating that ICG-001 is the most suitable inhibitor for further experiments (Fig. 6a-h). The MAML inhibitor IMR-1 showed strong effects both on spheres and organoids. In general, inhibition by IMR-1 was stronger than the effects observed by DAPT, specifically in organoids, yet it also had pronounced effects on normal cells (Fig. 6i-l).

10) Fig 6 – the effects of the Wnt/Notch inhibitors on the spheres/PDX tumors is potentially very interesting. However, I remain somewhat unconvinced of the selectivity of the drugs being used. Please include a non-cancer control. There also appears to be no nuclear b-catenin evident in the tumor samples as one might expect given the Wnt pathway activation status. As previously mentioned, Lgr4 might not be the most appropriate indicator of Wnt pathway status – please include a larger panel here. Is there apoptosis evident in the treated samples? Would Porcupine inhibitors work here? Why were PDX1 and PDX4 selected for this experiment in place of PDX1-3 as used in figure 1?

We have included results from normal sphere and organoid cultures for all inhibitors in the new version of the manuscript (please refer to Supplementary Figure 6).

We have added the following text on page 6 of the manuscript:

To exclude non-specific toxicity, we treated spheres and organoids derived from normal-adjacent tissue. DAPT had no significant effect on the 5 sphere and organoid cultures that were tested. ICG-001 produced no significant toxicity in sphere cultures, but led to a reduction of organoid growth.

To exclude that the effects of ICG-001 were due to unspecific events, we tested additional WNT inhibitors in specimens that responded to ICG-001. We used the TANKYRASE inhibitor XAV939⁵⁸ and the β -CATENIN inhibitor LF3⁵⁹, and the Porcupine inhibitor C59⁶⁰. While each of the WNT inhibitors reduced sphere formation and organoid growth, they showed different efficiencies, decreased the expression of WNT target genes to varying levels, and some had stronger effects on normal cells, indicating that ICG-001 is the most suitable inhibitor for further experiments (Fig. 6a-h). The MAML inhibitor IMR-1 showed strong effects both on spheres and organoids. In general, inhibition by IMR-1 was stronger than the effects observed by DAPT, specifically in organoids, yet it also had pronounced effects on normal cells (Fig. 6i-l).

DAPT does not have an effect in normal spheres or organoids, as well as ICG-001 in normal sphere cultures. Unfortunately, we observed an effect of ICG-001 in organoids, which might explain the stronger effect of ICG-001 in organoids of the same patient. However, in xenografts, we did not observe any adverse effect in the animals in the treatment period, as examined by weight of the animals, general well-being and macroscopic examination of the kidneys, the lung, the liver and the colon at the end of the treatment period. The chosen concentration of ICG-001 and DAPT have also been reported in previous studies to be tolerated in in vivo experiments (see Zhao et al. Sci Rep, 2016, doi:10.1038/srep24704 and Emami et al. PNAS, 2004, doi:10.1073/pnas.0404875101). We write on page 16:

In addition, we did not observe any signs of unspecific toxicity in mice at the end of the treatment period, as assessed by weight loss, general appearance and macroscopic examination of visceral organs. Both inhibitors have also been used in other preclinical models in the same or lower concentrations^{68,69}.

REVIEWERS' COMMENTS:

Reviewer #1 (Remarks to the Author):

In the revised version the authors have made substantial improvements of the manuscript and addressed most of my concerns. The authors have elaborated on the selection of the stem cell markers, and it is particularly rewarding that they have been staining primary tumors for the selected stem cell markers. These results are now well described and integrated in the manuscript. The weakness of the data presented in Figure 1E (now Figure 2b) regarding the limited number of mice transplanted with triple-sorted tumor cells remains, and still represents a weakness of the paper. In the response the authors acknowledge this point and refer to the laborious and time-consuming nature of these experiments. They should acknowledge this in the manuscript and point out that these results have to be interpreted with caution. In all, this study provides novel and interesting data regarding the nature of cancer stem cells in RCC that may aid in developing novel therapeutic modalities for treatment of the disease.

Reviewer #2 (Remarks to the Author):

The authors addressed my questions satisfactorily.

Reviewer #3 (Remarks to the Author):

I appreciate the efforts made to address my major critique - whilst a fair amount of written rebuttal (as oppose to experimental improvement) was included, I can agree to the majority. However, I still find the low numbers of xenograft samples used a potential problem with regards to reproducibility and I was disappointed with the lack of effort to validate some of the findings in primary human cancers. Whilst I sympathize with the technical difficulty of generating the xenograft models and obtaining primary samples for generating the biological replicates in the various assays, the limited sample sizes included do somewhat reduce the potential impact of the findings.

**Response to the reviewers
(Fendler et al. NCOMMS-19-15071A)**

Reviewer #1 (Remarks to the Author):

In the revised version the authors have made substantial improvements of the manuscript and addressed most of my concerns. The authors have elaborated on the selection of the stem cell markers, and it is particularly rewarding that they have been staining primary tumors for the selected stem cell markers. These results are now well described and integrated in the manuscript. The weakness of the data presented in Figure 1E (now Figure 2b) regarding the limited number of mice transplanted with triple-sorted tumor cells remains, and still represents a weakness of the paper. In the response the authors acknowledge this point and refer to the laborious and time-consuming nature of these experiments. They should acknowledge this in the manuscript and point out that these results have to be interpreted with caution. In all, this study provides novel and interesting data regarding the nature of cancer stem cells in RCC that may aid in developing novel therapeutic modalities for treatment of the disease.

Reviewer #2 (Remarks to the Author):

The authors addressed my questions satisfactorily.

Reviewer #3 (Remarks to the Author):

I appreciate the efforts made to address my major critique - whilst a fair amount of written rebuttal (as oppose to experimental improvement) was included, I can agree to the majority. However, I still find the low numbers of xenograft samples used a potential problem with regards to reproducibility and I was disappointed with the lack of effort to validate some of the findings in primary human cancers. Whilst I sympathize with the technical difficulty of generating the xenograft models and obtaining primary samples for generating the biological replicates in the various assays, the limited sample sizes included do somewhat reduce the potential impact of the findings.

We thank all reviewers for the positive response to our revision.

We have addressed the remaining concerns in the current version of the manuscript

Reviewer #1:

The weakness of the data presented in Figure 1E (now Figure 2b) regarding the limited number of mice transplanted with triple-sorted tumor cells remains, and still represents a weakness of the paper. In the response the authors acknowledge this point and refer to the laborious and time-consuming nature of these experiments. They should acknowledge this in the manuscript and point out that these results have to be interpreted with caution.

Reviewer #3:

However, I still find the low numbers of xenograft samples used a potential problem with regards to reproducibility and I was disappointed with the lack of effort to validate some of the findings in primary human cancers. Whilst I sympathize with the technical difficulty of generating the xenograft models and obtaining primary samples for generating the biological replicates in the various assays, the limited sample sizes included do somewhat reduce the potential impact of the findings..

We have added the following paragraph to the discussion to discuss the limitations regarding the xenograft assays (page 15, lines 12ff.):

A major limitation of this study is the low number of xenografts used to test tumor-initiating capacity. In our hands, xenografts of RCC grew slowly with three month latency until the formation of subcutaneous tumors, which limited expansion of these tumors. After sorting for CSCs, cell numbers were limited and did not allow for 3 technical replicates per concentration.